# Dysregulation of homeostatic cytokine receptors drives prolonged T cell activation following acute SARS-CoV-2 infection in humans

Laura Ceglarek [1,2,3], Patrick Taeschler [2], Alp Inci [1,2], Yves Zurbuchen [1,2], Sarah Adamo[2], Carlo Cervia-Hasler[2], Miro E. Raeber [1,2,3,4] & Onur Boyman [1,2,3,4,5] ✉

Acute viral infections are usually cleared by an efficient anti-pathogen immune response, following which immune homeostasis is restored. Occasionally, such pathogen-induced immune response fails to abate despite clinical recovery, but how this occurs in humans has not been thoroughly investigated. Here, we perform a detailed analysis of T cell homeostasis following severe acute respiratory syndrome coronavirus-2 (SARS-CoV-2) infection, which reveals persistent activation and dyshomeostasis of CD4[+] and CD8[+] T cells for 6–12 months after acute infection. Compared to steady-state and unlike T cell responses following vaccination, interleukin (IL)−2 receptor and IL-7 receptor expression remains altered on both SARS-CoV-2-specific and bystander T cells for 6–12 months after acute infection. These alterations correlate with increased IL-7 and IL-15 serum levels and are reproduced by in vitro stimulation by IL-7 and IL-15, but surprisingly not by IL-2. Collectively, our study demonstrates prolonged T cell dyshomeostasis driven by dysregulated homeostatic cytokine signals following acute viral infection.

Cytokines of the common gamma chain (γc; also termed CD132) receptor family are key to lymphocyte development and homeostasis. Signaling via γc cytokines facilitates the maintenance of stable lymphocyte counts during steady-state and following immune activation[1–3]. Interleukin (IL)−7 and IL-15 are essential for homeostatic survival and background proliferation of naive and memory CD4[+] and CD8[+] T cells, whereas IL-2 is crucial for the homeostasis of regulatory T (Treg) cells[2–4]. CD132 is abundantly expressed on T cells and is not limiting signaling capacity. In contrast, sensitivity to IL-2, IL-7 and IL-15 is mediated by a fine balance of constitutive versus demand-based, dynamic expression of their corresponding receptor subunits[1].

IL-2 can signal via two different IL-2 receptors (IL-2Rs), termed dimeric and trimeric IL-2Rs, both of which activate Janus kinase (JAK)−signal transducer and activator of transcription (STAT) pathways[4,5]. Dimeric IL-2Rs consist of two subunits comprising IL-2Rβ (CD122) and CD132, whereas the addition of IL-2Rα (CD25) to CD122 and CD132 forms trimeric IL-2Rs. CD25 does not contribute to signal transduction, but it increases receptor affinity by 10–100 fold[6]. At steady-state, high CD25 expression is found preferentially on thymus-derived forkhead box P3 (FOXP3)[+] Treg cells in humans, non-human primates, and mice[7,8]. Moreover, CD25 can be upregulated on activated conventional T cells, which allows these cells to compete with CD25[hi] Treg cells for

[1]Center for Human Immunology, University of Zurich, Zurich, Switzerland. [2]Department of Immunology, University Hospital Zurich, Zurich, Switzerland. [3]Department of Quantitative Biomedicine, University of Zurich, Zurich, Switzerland. [4]Faculty of Medicine, University of Zurich, Zurich, Switzerland. [5]Faculty of Science, University of Zurich, Zurich, Switzerland. ✉e-mail: onur.boyman@uzh.ch

limited IL-2 concentrations. Thus, acute viral infection in mice results in rapid CD25 upregulation on virus-specific T cells after infection, which is paralleled by an increase of CD122 and CD132 on these cells[9,10]. However, in mice, this upregulation is transient and returns to baseline levels within 1 week after antigen clearance.

The heterodimer of CD122 and CD132 not only forms the dimeric IL-2R but also the dimeric IL-15R[3,11]. At steady-state, IL-15 is produced in significant amounts by myeloid cells, which also express IL-15Rα. This allows them to efficiently trans-present IL-15Rα-bound IL-15 to cells expressing the dimeric IL-15R, including memory CD8$^+$ T and natural killer (NK) cells[12–14]. Both memory CD8$^+$ T cells and NK cells are very sensitive to and rely heavily on IL-15 signals for their homeostatic proliferation and survival. Moreover, memory CD8$^+$ T and NK cells are also very responsive to immunotherapy using IL-2 and CD122-biased IL-2 formulations, both in mice and humans[15–19].

Low-level IL-7 signals, transmitted via IL-7Rα (CD127) paired with CD132, are essential for homeostatic survival of naive and memory T cells[3,20–22]. CD127 expression is high on all T cell subsets at steady-state, except for FOXP3$^+$ Treg cells in humans[23,24]. Dynamic regulation of CD127 ensures that naive T cells gain access to limited IL-7 resources[25].

Most, if not all of our knowledge on dynamic regulation of IL-2Rs, IL-7Rs, and IL-15Rs in vivo is derived from mouse studies where mice were challenged with an acute viral infection. Even though the roles of these cytokines seem preserved from mice to humans, there are interspecies differences pertaining to the regulation of CD122 in NK and CD8$^+$ T cells or the role of IL-7 in B cell development[26–28]. Moreover, the results obtained in inbred mice might not capture the inter-individual variation in humans, both at steady-state and following acute infection. As different immunotherapies based on these cytokines are entering clinical trials[29,30], it is critical to obtain in vivo data in humans.

We have now addressed these outstanding issues by profiling the expression of IL-2R and IL-7R subunits on different lymphocyte subsets in humans during steady-state and following viral challenge, vaccination and IL-2 immunotherapy. To this end, we utilized our coronavirus disease 2019 (COVID-19) patient cohort[31–39], sampled longitudinally during acute infection with severe acute respiratory syndrome coronavirus 2 (SARS-CoV-2) and up to one year into recovery. Unexpectedly, we found that alterations in IL-2R and IL-7R surface abundance on T cells were not only restricted to the acute infection stage but also present up to 12 months following clinical recovery. This dysregulated receptor expression was particularly apparent in patients with severe disease and paralleled by chronic T cell activation. Using in vitro experiments and different human in vivo observations, we demonstrate that T cell receptor (TCR) signals resulted in notable transient changes of IL-2R and IL-7R expression in human T cells, whereas continuous cytokine signals by IL-7 and IL-15, but not IL-2, lead to more discrete sustained changes. In line with these findings, proteomic analysis of blood serum of individuals with persistently increased CD25 expression on T cells revealed sustained elevation of IL-7 and IL-15 levels. Thus, continuous and aberrant signaling by these cytokines likely contributes to a previously unappreciated long-lasting alteration of homeostatic cytokine signaling and chronic T cell activation after clinical recovery from acute SARS-CoV-2 infection. Therefore, accounting for an individual's immunological history, including infections resolved several months earlier, may be highly relevant for optimizing T cell-targeted immunotherapies and vaccination strategies.

## Results

### CD25, CD122, CD127 and CD132 are differentially expressed at steady-state

We assessed peripheral blood mononuclear cells (PBMCs) of 42 healthy individuals by spectral flow cytometry to characterize and quantify the surface abundance of CD25, CD122, CD127 and CD132 in 18 different lymphocyte subsets at steady-state (Fig. 1a and Supplementary Fig. 1a). Use of isotype-matched controls allowed us to obtain quantitative values of these cytokine receptor subunits, represented by the difference of geometric mean fluorescent intensity (ΔgMFI) of fully stained samples and isotype-matched controls (Fig. 1b and Supplementary Fig. 1b, c).

As previously reported[40], the highest CD25 expression was observed in Treg cells, which was more pronounced in activated (aTreg) compared to resting Treg (rTreg) cells (Fig. 1c). Within conventional CD4$^+$ (hereafter simply termed CD4$^+$) T cells, naive cells showed the lowest expression of CD25, followed by central-memory T (T$_{CM}$) and effector-memory T (T$_{EM}$) cells. In comparison to CD4$^+$ T cells, CD8$^+$ T cells exhibited lower CD25 levels in general, which were slightly elevated in T$_{CM}$ cells compared to naive, T$_{EM}$ and CD45RA$^+$ effector-memory T (T$_{EMRA}$) cells. As published[41], CD56$^{bright}$ NK cells showed higher CD25 levels than CD56$^{dim}$ NK cells. While naive and transitional B cells did not express CD25, intermediate levels of CD25 were detected on unswitched memory B cells (MBC), switched MBC, and CD27$^-$IgD$^-$ 'double-negative' (DN) B cells.

Among assessed populations, CD122 levels were highest in CD56$^{bright}$ NK cells, followed by CD56$^{dim}$ NK cells, CD8$^+$ T cells, CD4$^+$ T cells (Fig. 1d). CD122 expression on NK T cells was similar to that of CD8$^+$ T cells. CD122 was more abundant on aTreg compared to rTreg cells, suggesting an overall higher capacity for forming trimeric receptors in the former subset.

CD4$^+$ T cells exhibited the highest levels of CD127, with higher abundance on T$_{CM}$ and T$_{EM}$ compared to naive cells (Fig. 1e). In CD8$^+$ T cells, naive and T$_{CM}$ cells showed higher levels of CD127 compared to T$_{EM}$ and T$_{EMRA}$ cells. rTreg cells showed intermediate and CD56$^{bright}$ NK, and aTreg cells low to intermediate CD127 expression. Very low to undetectable levels of CD127 were observed in CD56$^{dim}$ NK cells and in all subsets of the B cell lineage.

CD132 was ubiquitously present in almost all assessed lymphocyte subsets, with subtle differences between these subsets (Fig. 1f). Memory T cells showed higher CD132 abundance than their naive counterparts, except for CD8$^+$ T$_{EMRA}$ cells. Also, aTregs and CD56$^{bright}$ NK cells showed slightly higher CD132 levels compared to rTregs and CD56$^{dim}$ NK cells, respectively. Overall, apart from transitional B cells, plasmablasts and plasma cells, all assessed lymphocyte subsets showed robust CD132 expression, suggesting that CD132 did not serve as a cytokine signal-modulating subunit in these lymphocytes.

### SARS-CoV-2 booster vaccination induces transient IL-2R and IL-7R changes

IL-2R and IL-7R expression is known to be predominantly affected by TCR signals. In order to characterize dynamic changes of cytokine receptor levels following antigenic stimulation in human T cells in vivo, we investigated a SARS-CoV-2 booster vaccination cohort as a model of antigen-driven T cell activation with minimal systemic inflammatory response (Supplementary Fig. 2). We assessed cytokine receptors on SARS-CoV-2-specific CD4$^+$ and CD8$^+$ T cells by using major histocompatibility complex (MHC) dextramers (Dex) to detect SARS-CoV-2 spike-specific (CoV2−Dex$^+$) T cells at steady-state prior to vaccination (pre-Vac) as well as 3 days (d), 10 d, and 30 d following vaccination (post-Vac) with the mRNA vaccine BNT162b2 (Fig. 2a and Supplementary Fig. 3a). We observed a robust increase in CoV2−Dex$^+$ CD4$^+$ and CD8$^+$ T cells following immunization, with CoV2−Dex$^+$ T cell frequencies reaching a maximum at 10 d and remaining elevated at 30 d post-Vac (Fig. 2b). Changes in IL-2R and IL-7R after vaccination were restricted to the CoV2−Dex$^+$ T cell compartment, whereas no changes were observed in CoV2−Dex$^-$ CD4$^+$ and CD8$^+$ T cells. CoV2−Dex$^+$ cells showed a significant upregulation of CD25 and CD122 3 d post-Vac (Fig. 2c). While elevated CD122 levels were maintained for more than

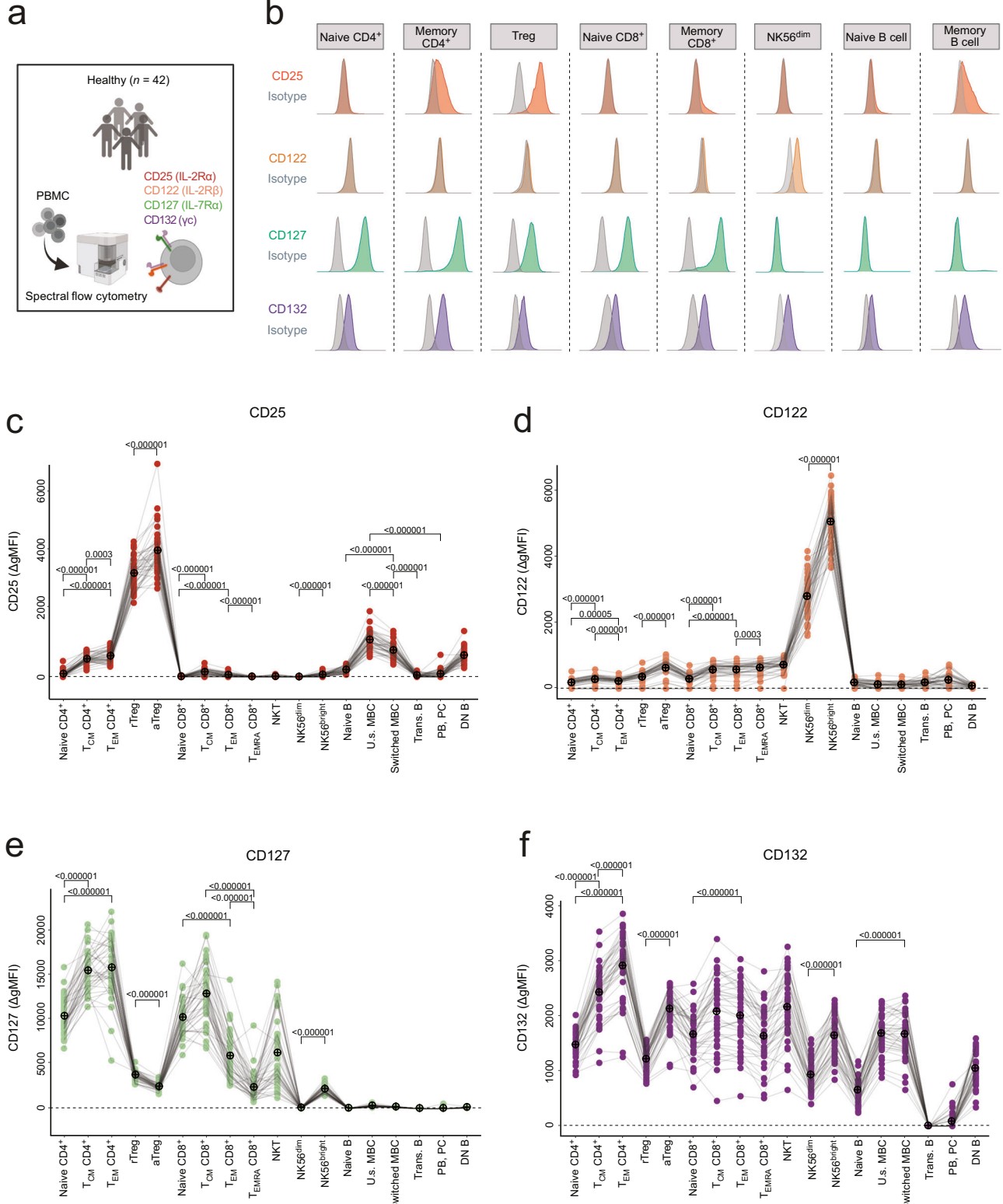

**Fig. 1 | Interleukin (IL)-2 and IL-7 receptor abundance on lymphocyte subsets at steady-state. a** Experimental approach for quantification of IL-2 receptor α (IL-2Rα; CD25), IL-2/IL-15Rβ (CD122), IL-7Rα (CD127) and γc (CD132) by spectral flow cytometry on peripheral lymphocyte subsets of healthy individuals (n = 42). Created in BioRender. Boyman, O. (2025) https://BioRender.com/fy1eeuq. **b** Histograms of CD25, CD122, CD127 and CD132 compared to isotype controls. Geometric mean fluorescence intensity (gMFI) difference of CD25 (**c**), CD122 (**d**) CD127 (**e**) and CD132 (**f**), comparing stained samples to isotype controls (ΔgMFI). Black crossed circles indicate mean values. Gray lines connect data points of same

individuals (n = 42). aTreg activated regulatory T, DN B CD27⁻IgD⁻ double-negative B, MBC memory B, NK natural killer, NK56^bright CD56^bright natural killer, NK56^dim, CD56^dim natural killer; NKT natural killer T; PB plasmablast; PC plasma cell, rTreg resting regulatory T, $T_{CM}$ central-memory T, $T_{EM}$ effector-memory T, $T_{EMRA}$ CD45RA⁺ effector-memory T, Trans. B transitional B, U.s. MBC unswitched memory B. For c–f, p values were determined using a paired, two-sided Wilcoxon signed-rank test and were adjusted for multiple comparisons using the Benjamini-Hochberg method. Source data are provided in the Source Data file.

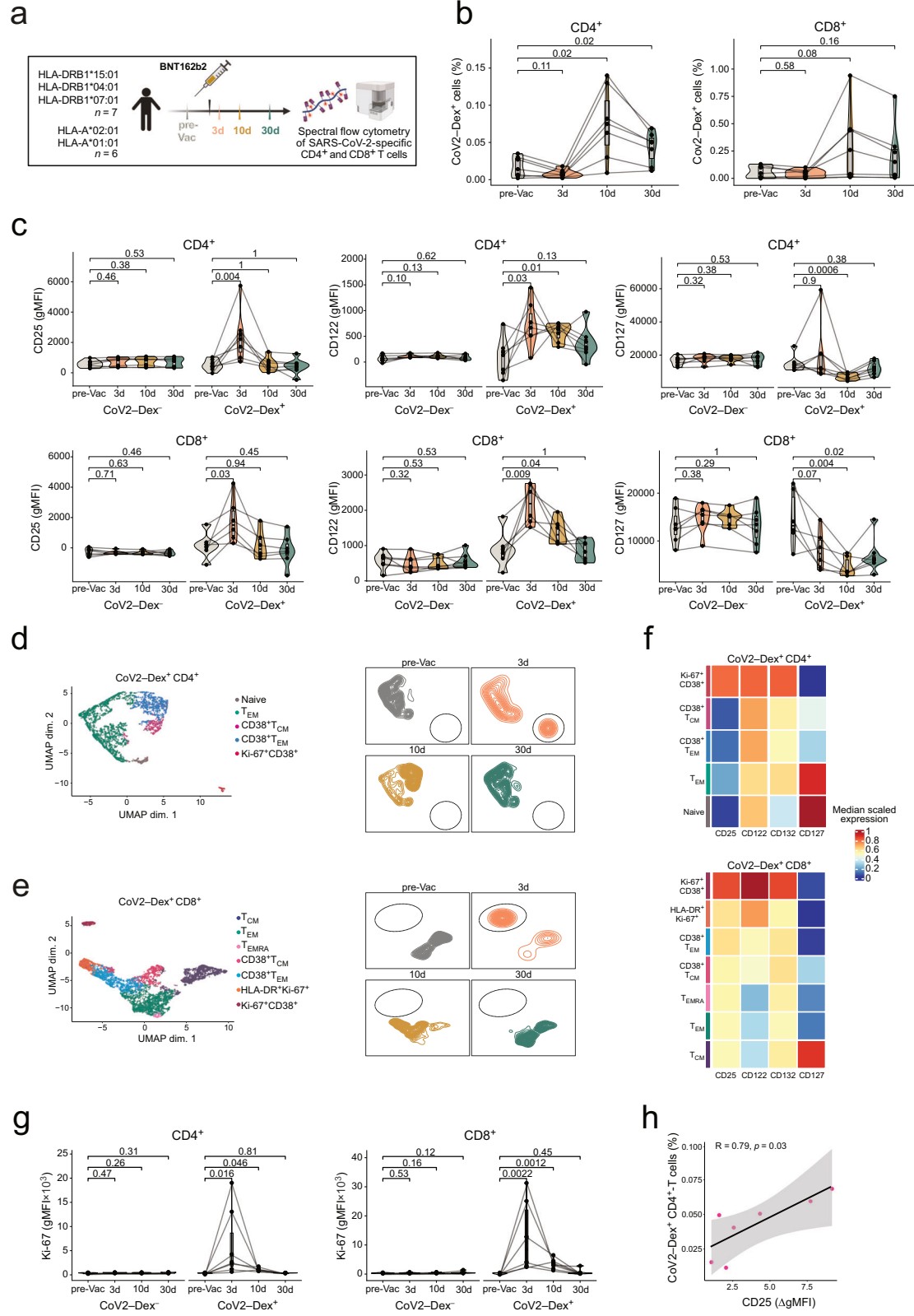

10 d post-Vac, the increase of CD25 was less sustained. In contrast, CD127 levels were reduced after immunization, reaching trough levels at 10 d post-Vac in both CD4[+] and CD8[+] T cells. This downregulation was more pronounced than the changes of the other receptor subunits, particularly in CoV2–Dex[+] CD8[+] T cells at 30 d post-Vac (Fig. 2c).

**Upon immunization CD25 precedes T cell activation markers and is associated with proliferation**

To investigate the distribution of IL-2R and IL-7R expression in the CoV2–Dex[+] compartment, we characterized the phenotypes of CoV2–Dex[+] T cells during immunization and identified different clusters (Fig. 2d, e, and Supplementary Fig. 3b). Phenotypically,

**Fig. 2 | Characterization of IL-2R and IL-7R on SARS-CoV-2-specific T cells after vaccination. a** Vaccination cohort overview. Created in BioRender. Boyman, O. (2025) https://BioRender.com/58qpoa1. **b** Frequencies of circulating severe acute respiratory syndrome coronavirus-2 (SARS-CoV-2) dextramer (CoV2−Dex)$^+$ CD4$^+$ (left, $n = 6$ healthy individuals) and CD8$^+$ (right, $n = 7$ healthy individuals) T cells following BNT162b2 booster vaccination. Lines connect data points of same individuals. **c** gMFI of CD25, CD122 and CD127 on CoV2−Dex$^+$ and CoV2−Dex$^-$ CD4$^+$ (top, $n = 6$) and CD8$^+$ T cells (bottom, $n = 7$) following booster vaccination. Lines connect data points of same individuals. Left, uniform manifold approximation and projection (UMAP) plot of gated CoV2−Dex$^+$ CD4$^+$ (**d**) and CD8$^+$ (**e**) T cell subsets of pre-vaccination ($n = 7$), 3 d ($n = 7$), 10 d ($n = 7$) and 30 d post-vaccination ($n = 7$) samples. Right, UMAP density plots of CoV2−Dex$^+$ CD4$^+$ and CD8$^+$ T cells at different timepoints pre- and post-vaccination. UMAP regions corresponding to CD25 expressing cell clusters are circled. **f** Heatmap of scaled marker expression in different CoV2−Dex$^+$ CD4$^+$ and CD8$^+$ T cell subsets across all timepoints. **g** gMFI of Ki-67 on CoV2−Dex$^+$ CD4$^+$ (left, $n = 6$ healthy individuals) and CoV2−Dex$^+$ CD8$^+$ T cells (right, $n = 7$ healthy individuals) following booster vaccination. Lines connect data points of same individuals. **h** Correlation of CD25 induction 3 d post-vaccination and end-point frequency of CoV2−Dex$^+$ CD4$^+$ T cells 30 d post-vaccination ($n = 7$). Correlation of numerical variables was determined by two-tailed Spearman's rank test. Gray band indicates 95% confidence interval. For comparison of cell subset frequencies or marker expression, timepoints were compared using a paired, two-sided Wilcoxon signed-rank test. $P$ values were adjusted for multiple comparisons using the Benjamini-Hochberg method. Box-plots are centered around the median; lower and upper hinges correspond to first and third quartiles, respectively; and lower and upper whiskers extend from hinge to smallest and largest values no further than 1.5× the interquartile range (IQR), respectively. Source data are provided in the Source Data file.

CoV2−Dex$^+$ cells transitioned from naive and resting memory phenotypes at steady-state to an activated effector phenotype 10 d post-Vac, which was characterized by expression of several T cell activation markers, such as CD38, programmed cell death-1 (PD-1), human leukocyte antigen (HLA)-DR, and inducible costimulator (ICOS) (Fig. 2d, e, and Supplementary Fig. 3c, d). Interestingly, elevated CD25 and CD122 expression was temporally separated from this activated phenotype and associated with a very distinct Ki-67$^+$ CD38$^+$ cluster (Fig. 2f, g), exclusively present at 3 d post-Vac. This cluster was characterized by high levels of proliferation, activation, and—in the case of CD8$^+$ cells— cytotoxicity. Despite its activated phenotype, HLA-DR expression was absent from the Ki-67$^+$ CD38$^+$ CD25$^+$ cluster and only emerged 10 d post-Vac (Supplementary Fig. 3e).

These findings indicated early and transient expression of the trimeric IL-2R after antigen stimulation. Increased IL-2 sensitivity could confer additional activation and proliferation queues leading to a more robust establishment of T cell memory. Indeed, increased CD25 levels on CoV2−Dex$^+$ CD4$^+$ T cells 3 d post-Vac correlated positively with activation at later timepoints and with increased frequencies of total detectable CoV2−Dex$^+$ CD4$^+$ T cells 30 d post-Vac (Fig. 2h and Supplementary Fig. 3f). By contrast, CD25 upregulation on CoV2−Dex$^+$ CD8$^+$ T cells was not correlated with percentages of CoV2−Dex$^+$ CD8$^+$ T cells, but rather with an increase in effector phenotype characterized by high T-box expressed in T cells (T-bet) expression and cytotoxicity (Supplementary Fig. 3f–h).

## Sustained changes in IL-2R and IL-7R abundance after acute SARS-CoV-2 infection

To investigate whether the trajectories of IL-2R and IL7-R expression during vaccination are recapitulated in the context of an acute viral infection, we compared these receptors in 25 healthy controls and 64 individuals with acute SARS-CoV-2 infection (Fig. 3a). Clinical manifestations associated with acute SARS-CoV-2 infection ranged from mild to severe COVID-19. No significant difference in age, sex, or comorbidities was apparent in infected individuals compared to healthy controls (Supplementary Table 1). Profiling IL-2R and IL-7R subunit abundance of healthy controls and COVID-19 patients during acute infection (Acute), 6 months (6 M follow-up) and 12 months after clinical recovery (12 M follow-up) revealed prolonged changes in these receptors, most notably in the T cell compartment. Compared to healthy controls, T cells of acutely infected individuals showed significantly higher levels of CD25 and CD122 (Fig. 3b). Increased CD25 levels on T cells strongly correlated with elevated serum concentrations of soluble CD25 (sIL-2Rα) during acute COVID-19 (Supplementary Fig. 4a), suggesting that CD25 was shed from the cells' surface by proteolytic cleavage[4]. This release of sIL-2Rα was furthermore increased in patients with severe COVID-19 (Supplementary Fig. 4b).

Unexpectedly, we found that both CD4$^+$ and CD8$^+$ T cells still carried significantly increased CD25 levels at 6 M follow-up (Fig. 3c), in stark contrast to the transient receptor alteration observed in the vaccination cohort. Similarly, 6 months after recovery, CD127 levels on both T cell subsets were strongly reduced. Most of these alterations in receptor expression returned to normal levels only by 12 months after recovery (Fig. 3c and Supplementary Fig. 4c). Paired analysis of CD25 levels in COVID-19 patients revealed unaltered CD25 expression on CD4$^+$ T cells and only a slight reduction of CD25 on CD8$^+$ T cells at 6 M follow-up compared to acute disease. However, CD25 levels on CD4$^+$ and CD8$^+$ T cells decreased significantly at 12 M follow-up. Similarly, CD122 abundance on CD4$^+$ and CD8$^+$ T cells remained high until 6 months after acute infection and declined to normal ranges only between six to 12 months. Conversely, CD127 expression exhibited a marked reduction at 6 M follow-up, which was normalized only at 12 M follow-up. While these persistent receptor alterations were observed in all COVID-19 patients, they were especially pronounced in individuals exhibiting severe disease.

Since individual memory T cell subsets express different levels of IL-2R and IL-7R (Fig. 1), we examined whether the overall change in receptor abundance in the T cell compartment was caused by a change in the relative frequencies of different memory T cell subsets. After acute COVID-19, we observed a relative increase in T$_{EM}$ cells at 6 M follow-up, in both CD4$^+$ and CD8$^+$ T cells, which was paralleled by a concomitant decrease in naive T cell frequencies (Supplementary Fig. 4d, e) as previously reported[39]. Naive T cell frequencies returned to homeostatic levels between 6 M and 12 M follow-up. However, the changes in IL-2R and IL-7R expression in T cells were not associated with a distinct T cell subset and similar in each subset assessed (Supplementary Fig. 5a). Interestingly, also naive T cells displayed similar receptor dynamics as memory T cells, suggesting that cytokines, rather than antigen receptor signals, mediated the protracted changes in homeostatic cytokine receptors.

## CD25 levels on T cells correlate with inflammatory environment and prolonged T cell activation

The increase in CD25 levels was particularly apparent and sustained in patients exhibiting severe infection, thus suggesting a contribution of an increased inflammatory environment to these receptor alterations. During acute COVID-19, CD25 levels were positively associated with several serum inflammation markers, including C-reactive protein (CRP), IL-1β, IL-6, IL-10, and tumor necrosis factor (TNF) (Fig. 3d). Furthermore, CD25 levels correlated with HLA-DR expression on CD4$^+$ and CD8$^+$ T cells during acute SARS-CoV-2 infection (Fig. 3d). CD122 expression was not associated with serum inflammation markers but correlated significantly with levels of HLA-DR on CD4$^+$ and CD8$^+$ T cells (Fig. 3d). Contrarily, the abundance of CD127 on CD8$^+$ T cells correlated negatively with HLA-DR expression in CD8$^+$ T cells. As previously described[38], HLA-DR expression on CD4$^+$ and CD8$^+$ T cells was higher in patients during acute SARS-CoV-2 infection

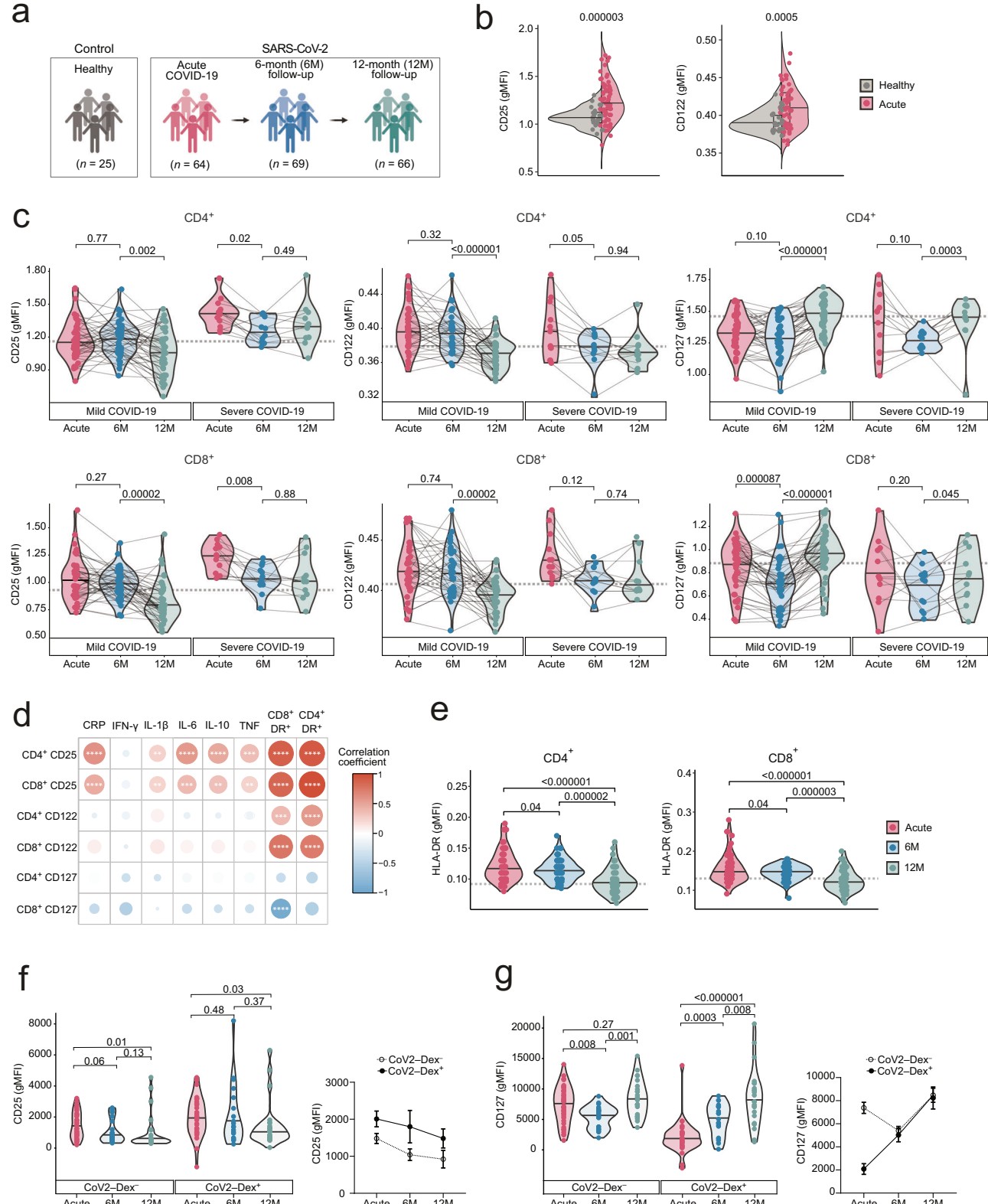

compared to healthy individuals. However, surprisingly, HLA-DR also remained elevated on CD4$^+$ and CD8$^+$ T cells at 6 M after acute infection, compared to healthy controls, and decreased between 6 M and 12 M follow-up (Fig. 3e), indicating the presence of activated T cells up to 6 months after acute SARS-CoV-2 infection.

**Bystander and TCR activation differently affect γc receptor expression on T cells**

To delineate the differential effects of TCR- versus cytokine-mediated signals on IL-2R and IL-7R expression, we profiled antigen-specific CD8$^+$ T cells by using fluorescently-labeled CoV2–Dex (Fig. 3f, g)[31]. Overall, CD25 expression followed a similar trajectory in CoV2–Dex$^-$

**Fig. 3 | CD25, CD122 and CD127 expression following acute SARS-CoV-2 infection. a** Cohort overview. Created in BioRender. Boyman, O. (2025) https://BioRender.com/e35qolj. **b** CD25 and CD122 abundance on T cells of healthy controls (*n* = 25) and COVID-19 patients during acute SARS-CoV-2 infection (*n* = 64). Statistical analysis by two-tailed t-test. Boxplots are centered around the median; lower and upper hinges correspond to first and third quartiles, respectively; and lower and upper whiskers extend from hinge to smallest and largest values no further than 1.5× the IQR, respectively. **c** CD25, CD122, and CD127 on CD4+ and CD8+ T cells during acute SARS-CoV-2 infection, and at 6-month (6 M) and 12-month (12 M) follow-up in patients exhibiting mild (*n* = 42) or severe disease (*n* = 11). Horizontal dashed lines indicate median of healthy controls (*n* = 25). **d** Matrix showing correlation of CD25, CD122 and CD127 gMFIs on CD4+ and CD8+ T cells with serum inflammation and T cell activation markers during acute COVID-19

(*n* = 64). Colored bar indicates different R values based on two-tailed Spearman's rank correlation coefficient. **e** HLA-DR expression in CD4+ and CD8+ T cells during acute SARS-CoV-2 infection (*n* = 64), and at 6 M (*n* = 69) and 12 M (*n* = 66) follow-up. Horizontal dashed lines indicate median of healthy controls (*n* = 25). CD25 (**f**) and CD127 (**g**) abundance in CoV-2–Dex+ and CoV-2–Dex– CD8+ T cells of COVID-19 patients during acute SARS-CoV-2 infection (*n* = 27), and at 6 M (*n* = 21) and 12 M follow-up (*n* = 29). Individuals with less than five CoV-2–Dex+ T cells were excluded from this analysis. Symbols indicate median [± standard deviation (SD)] CD25 and CD127 levels of CoV-2–Dex+ and CoV-2–Dex– CD8+ T cells. * *p* < 0.05; **\**p* < 0.01; ***\**p* < 0.001; **** *p* < 0.0001. Comparison of marker expression throughout time was done using a paired, two-sided Wilcoxon signed-rank test. *P* values were adjusted for multiple comparisons using the Benjamini-Hochberg method. Source data are provided in the Source Data file.

and CoV2–Dex+ CD8+ T cells, with a marked increase of CD25 during acute infection, which partially persisted at 6 M follow-up and decreased at 12 M follow-up (Fig. 3f). However, a tendency of more pronounced CD25 upregulation was noted in the CoV2–Dex+ T cell compartment, with some individuals exhibiting particularly high levels at 6 M follow-up. The maintenance of elevated CD25 levels at 6M-follow-up could be caused by lingering antigen or SARS-CoV2 viral reservoirs[42,43]. To investigate this possibility, we assessed the phenotype of CoV2–Dex+ T cells with regard to effector T cell markers (Supplementary Fig. 5b). As previously reported[31], an effector signature characterized by high HLA-DR, Ki-67, CXCR3, and the transcription factors eomesodermin and T-bet was observed in CoV2–Dex+ during acute COVID-19. This effector signature decreased at 6 M and 12 M follow-up, suggesting that remaining CoV2–Dex+ T cells represented memory T cells.

Compared to CoV2–Dex+ T cells, CD25 levels on CoV2–Dex– were consistently lower. This disparity in receptor dynamics was even more apparent in CD127 expression (Fig. 3g). CoV2–Dex+ CD8+ T cells showed a drastic downregulation of CD127 during acute infection, which partially persisted at 6 M follow-up, whereas CoV2–Dex– cells only showed a mild decrease of CD127 levels at 6 M follow-up. CD127 levels subsequently returned to baseline levels between 6 M and 12 M follow-up in both CoV2–Dex+ and CoV2–Dex– CD8+ T cells. Similar to bulk T cells, both naive and memory CoV2–Dex– cells displayed similar kinetics in CD25 and CD127 expression and showed increased proliferation at 6 M follow-up (Supplementary Fig. 5c).

Since CoV2–Dex– cells include both T cells specific to other SARS-CoV-2 epitopes not included in our dextramer reagents and to unrelated targets, we assessed whether memory CD8+ T cells specific to other viruses showed protracted changes of their IL-2Rs and IL-7Rs. To this end, we longitudinally profiled Epstein-Barr virus (EBV)-specific and influenza A virus (IAV)-specific CD8+ T cells during acute COVID-19, 6 M and 12 M follow-up (Fig. 4a, and Supplementary Fig. 6a). COVID-19 patients showed comparable levels of EBV- and IAV-specific CD8+ T cells compared to healthy controls (Fig. 4b). CoV2–Dex+ cells showed significantly elevated CD25 and CD122 levels with a concomitant decrease in CD127 during acute infection (Fig. 4c). Notably, also EBV- and IAV-specific CD8+ T cells showed increased CD25 and CD122 during acute COVID-19, which was still present at 6 M follow-up and only returned to healthy levels at 12 M follow-up (Fig. 4d). IAV-specific T cells showed a mild decrease in CD127, which reverted to basal levels at 12 M follow-up. By contrast, EBV-specific T cells showed persistently low CD127, which possibly reflected their high percentages of T_EM and T_EMRA phenotypes (Supplementary Fig. 6b).

To investigate whether differential IL-2R and IL-7R expression on bystander T cells correlated with cellular activation, we integrated the data of all antigen-specific T cell subsets and performed unbiased clustering (Fig. 4e). SARS-CoV-2-, IAV-, and EBV-specific T cells exhibited a predominant T_EM phenotype, with SARS-CoV-2- and EBV-specific T cells showing a sizable proportion of T_EMRA cells (Fig. 4e and

Supplementary Fig. 6b). During acute COVID-19, cell clusters emerged co-expressing several T cell activation markers including HLA-DR, granzyme B, and CD38 (Supplementary Fig. 6c). Particularly, the proliferating and highly activated T_EM cluster (Activ. prolif. T_EM) exhibited high CD25 and CD122 and a concomitant decrease in CD127 (Fig. 4f and Supplementary Fig. 6c–e). On IAV-specific CD8+ T cells, CD122 upregulation and CD127 downregulation correlated significantly with the cells' activation status across all sampling timepoints (Fig. 4g, h). In line with their changes in IL-2Rs and IL-7Rs, IAV-specific T cells showed increased HLA-DR, T-bet, and Ki-67, which only returned to healthy steady-state at 12 M follow-up (Fig. 4i). Collectively, these data indicated prolonged alterations of IL-2R and IL-7R expression in T cells after acute SARS-CoV-2 infection, which persisted for 6 months after clinical recovery. These changes were driven by antigen-dependent and inflammatory signals and were most prominent in individuals experiencing severe disease.

## CD25 expression is increased in individuals with persistent post-acute infection syndrome

To investigate whether persistent T cell activation following acute infection correlated with post-acute infection syndrome, we compared receptor levels of individuals that clinically recovered from COVID-19 to those who reported long COVID (LC) symptoms for up to 12 months after acute infection (Supplementary Fig. 7a). Total T cells showed higher CD25 expression levels at 12 M follow-up in individuals exhibiting LC symptoms at this timepoint (Supplementary Fig. 7a), suggesting that persistent T cell activation by enhanced cytokine sensing could contribute to low-grade chronic inflammation. Furthermore, increased CD25 expression was prominent in individuals exhibiting multiple LC symptoms (Supplementary Fig. 7b). Since we demonstrated that the magnitude of receptor alterations is dependent on initial disease severity (Fig. 3c), we compared CD25 levels on polyclonal T cells (Supplementary Fig. 7c) and on SARS-CoV-2- and IAV-specific CD8+ T cells (Supplementary Fig. 7d) of a subset of individuals who all experienced a mild disease course[34]. CD25 levels were elevated in SARS-CoV-2-specific CD8+ T cells of LC patients at 6 M follow-up, whereas IAV-specific CD8+ T cells only showed a tendency of increased CD25 expression. Furthermore, SARS-CoV-2-specific CD8+ T cells of LC patients with active symptoms 12 months after acute disease showed higher CD25 expression compared to convalescent individuals, unlike their IAV-specific CD8+ T cell counterparts (Supplementary Fig. 7d).

## Increased serum IL-7 and IL-15 are associated with sustained IL-2R and IL-7R changes

To determine whether sustained cytokine signaling could be the reason for prolonged cytokine receptor alteration following COVID-19, we investigated the serum proteome of our COVID-19 cohort. Indeed, we found significantly higher concentrations of IL-7 and IL-15 in patients with acute severe disease, and increased concentrations of IL-15, but not of IL-7, were still present at 6 M follow-up (Fig. 5a and

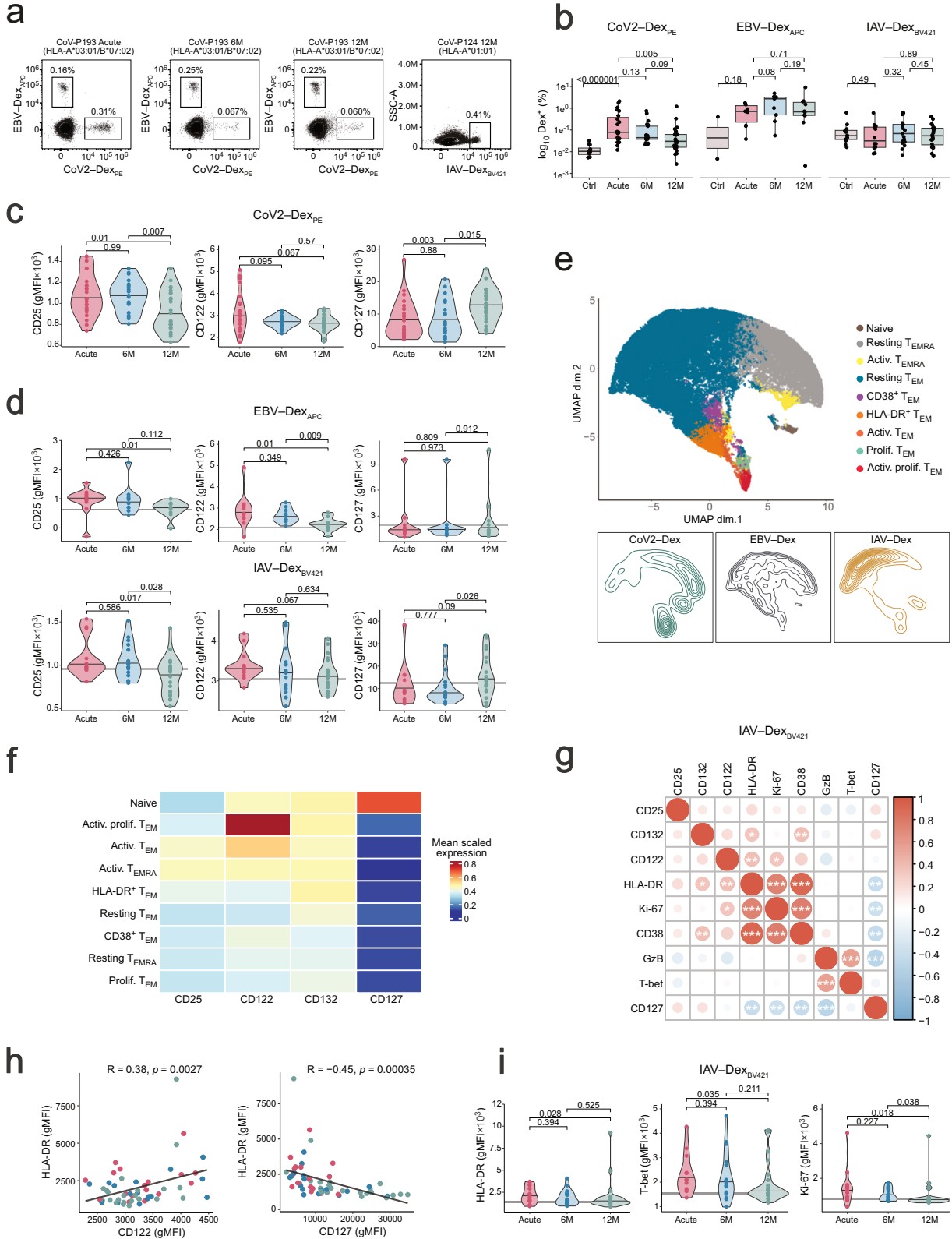

As T cells compete for limited availability of homeostatic cytokines, depletion of T cells during acute infection may lead to increased concentrations of IL-7 and IL-15, as previously shown for IL-7 during acute COVID-19[39]. In line with this observation, high CD25 levels positively correlated with IL-15 levels during acute infection (Fig. 5b and Supplementary Fig. 8c).

Since IL-15 is presented in cell-bound form by IL-15Rα-expressing cells and substantial amounts of IL-7 are acting locally in tissues and are not released into circulation, total T cell counts may serve as a more robust surrogate metric than serum cytokine levels for estimating the effective IL-7 and IL-15 concentrations acting on lymphocytes. As previously reported[39], T cell counts during acute disease were reduced

**Fig. 4 | Durable changes of γc receptor subunits on bystander T cells upon acute infection. a** Representative flow cytometry plots of SARS-CoV-2-, Epstein-Barr virus (EBV)-, and influenza A virus (IAV)-specific CD8[+] T cells, identified by peptide-loaded MHC-dextramers (CoV2–Dex, EBV–Dex, and IAV–Dex, respectively), during acute COVID-19, and at 6 M and 12 M follow-up. **b** Frequencies of antigen-specific CD8[+] T cells identified by indicated dextramers in healthy individuals (Ctrl; n = 13) as well as in COVID-19 patients (n = 26) at indicated timepoints. Statistical analysis by two-sided Wilcoxon signed-rank test. CD25, CD122 and CD127 expression on CoV2–Dex[+] CD8[+] T cells (**c**) and EBV- and IAV-specific CD8[+] T cells (**d**) in COVID-19 patients (n = 26) at indicated timepoints. Subjects with less than five detectable Dex[+] T cells were excluded. Gray lines indicate receptor levels on corresponding cells in healthy controls (n = 13). In **b**–**d** HLA restriction prevented use of certain dextramers in all subjects. **e** UMAP projection of CoV2-, IAV- and EBV-specific CD8[+] T cells in healthy individuals and COVID-19 patients, clustered based on activation markers and split by antigen specificity. **f** Heatmap depicting mean expression of indicated receptor subunits in different clusters of CoV2-, EBV- and IAV-specific CD8[+] T cells. **g** Correlation between indicated receptor subunits and markers on IAV-specific CD8[+] T cells. **h** Correlation of HLA-DR and CD122 and CD127 expression on IAV-specific CD8[+] T cells, depicted as overall trend (black line), as well as data points during acute infection (magenta; n = 17), and at 6 M (blue; n = 17) and 12 M follow-up (green; n = 17). For g and h, correlation is indicated by Pearson's correlation coefficient. **i** HLA-DR, T-bet and Ki-67 expression on IAV-specific CD8[+] T cells in COVID-19 patients (n = 17) at indicated timepoints. *p < 0.05; **p < 0.01; ***p < 0.001; ****p < 0.0001. For **c**, **d**, and **i**, p values were determined by paired, two-sided Wilcoxon signed-rank test and adjusted for multiple comparisons using the Benjamini-Hochberg method. Boxplots are centered around the median; lower and upper hinges correspond to first and third quartiles, respectively; and lower and upper whiskers extend from hinge to smallest and largest values no further than 1.5× the IQR, respectively. Source data are provided in the Source Data file.

compared to healthy steady-state, remained slightly decreased at 6 M follow-up and only recovered to steady-state levels at 12 M follow-up (Supplementary Fig. 8d). This decrease in T cell counts could in part be due to migration of effector T cells to inflamed tissues but also indicate cell death and lymphopenia followed by a gradual reconstitution. CD4[+] and CD8[+] T cell counts inversely correlated with serum IL-15 levels (Fig. 5c and Supplementary Fig. 8e). Moreover, across all three sampling timepoints, overall CD25 expression by T cells inversely correlated with T cell counts, which were consistently reduced in patients exhibiting severe disease (Fig. 5d). Collectively, these data suggest a mechanism of prolonged, lymphopenia-driven increase of systemic IL-7 and IL-15 concentrations after acute SARS-CoV-2 infection, which sustains increased IL-2R and decreased IL-7R expression.

## IL-7 and IL-15 signaling modulates IL-2R and IL-7R expression

To mechanistically confirm the contribution of elevated serum IL-7 and IL-15 signals to maintaining altered receptor profiles, we performed in vitro T cell stimulation experiments. Expression of CD25, CD122, CD132 and CD127 was assessed in magnetic-activated cell sorting (MACS)-enriched CD4[+] and CD8[+] T cells after stimulation in vitro with different cytokines (Fig. 6a, b, and Supplementary Figs. 9 and 10a, b). IL-7 and IL-15 stimulation resulted in upregulation of CD25, whereas IL-2, IL-4, IL-7, and IL-15 reduced CD127 levels (Fig. 6a, b). We did not find significant alterations after stimulation with IL-21, IL-6, IFN-γ, and TNF, and levels of CD122 and CD132 were unaffected by all cytokines tested.

Subsequently, we characterized cytokine-mediated induction of CD25 in more detail by assessing the response of different memory T cell subsets to γc cytokine stimulation (Fig. 6c and Supplementary Fig. 10c, d). Both IL-7 and IL-15 resulted in a dose-dependent increase in CD25 expression on memory CD4[+] and CD8[+] T cells. Surprisingly, IL-2 did not alter CD25 expression in these memory T cell subsets during in vitro stimulation for 48 h. By contrast, Treg cells exhibited a subtle CD25 upregulation already at intermediate IL-2 doses (Fig. 6c and Supplementary Fig. 10e). While IL-2 and IL-15 resulted in differential modulation of CD25 on memory T cells, Foxp3 upregulation and CD127 downregulation were comparable after IL-2 and IL-15 treatment (Fig. 6d and Supplementary Fig. 10f–h). IL-7 mediated the strongest CD127 downregulation on all assessed T cell subsets, as previously described[25]. Reflecting the finely graded IL–2R and IL-7R expression levels on different T cell subsets (Fig. 1f), the magnitude of CD25 and CD127 modulation was distinctive. Whereas memory CD8[+] T cells responded strongly to IL-15 but only weakly to IL-7, CD4[+] T cells were equally responsive to IL-15 and IL-7, the latter likely mediated by their high CD127 expression (Fig. 6e–g). Having identified IL-7 and IL-15 as the dominant cytokines able to increase CD25 and decrease CD127 levels in the absence of antigenic stimulation, we stimulated T cells either continuously for 96 h or treated them with cytokines for 24 h,

followed by removal of cytokines (Supplementary Fig. 11a–d). Continuous stimulation led to a successive increase in CD25 on memory T cells. Conversely, CD25 levels tended to decrease upon removal of the cytokine stimulus. Similarly, IL-7- and IL-15-mediated downregulation of CD127 was sustained by continuous stimulation, whereas removal of stimulation led to partial recovery (Supplementary Fig. 11d). Increasing the exposure time to cytokine to 48 h led to a more sustained upregulation of CD25 suggesting that prolonged cytokine signals not only resulted in an increased magnitude of receptor modulation but also to a more persistent consolidation of altered receptor levels (Supplementary Fig. 11e, f). Thus, these experiments indicated that sustained IL-7 or IL-15 signals, but not IL-2, were required to maintain altered IL-2R and IL-7R levels.

Since surface abundance of γc receptors is modulated by extracellular and intracellular pools of receptor subunits, the latter of which can be externalized to the cell surface, we investigated whether cytokine stimulation alters the total receptor pool. To this end, we compared the increase in CD25 on the cell surface to the increase in total (surface and intracellular) receptor expression (Fig. 6g and Supplementary Fig. 12a). On both memory CD4[+] and CD8[+] T cells, surface CD25 showed a dose-dependent increase after stimulation by IL-7 or IL-15. By contrast, total CD25 levels did not increase to the same extent. Thus, the discrepancy in CD25 surface levels after IL-2 and IL-15 stimulation was not due to increased CD25 internalization after stimulation with IL-2. To exclude obstruction of the antibody epitope by IL-2 binding to the IL-2R, we tested CD25 levels in prestimulated T cells with two anti-CD25 antibody clones, in presence and absence of high doses of IL-2, revealing no significant decrease of CD25 by the presence of IL-2 (Supplementary Fig. 12b). Furthermore, signals of both anti-CD25 antibody clones showed similar dose-response profiles to stimulation with IL-2, which was consistently lower than the CD25 signal induced by IL-7 stimulation at equal doses (Supplementary Fig. 12c, d). We further confirmed these findings by using an IL–2–antibody complex (termed NARA1leukin)[18], which blocks the CD25 epitope of IL-2; thus, CD25 induced by NARA1leukin stimulation is unlikely to be masked by ligand binding to the trimeric IL-2R. However, stimulation with equimolar concentrations of NARA1leukin did not lead to more pronounced upregulation of CD25 (Supplementary Fig. 12e, f). Lastly, to confirm the observed changes in receptor levels were mediated by direct γc cytokine signaling, we abrogated JAK activity by using the JAK1/3 inhibitor Tofacitinib (Fig. 6h). To account for the partial decrease in CD127 signal in the presence of IL-7, most likely due to partial occlusion of the anti-CD127 antibody epitope by IL-7, we calculated fold differences in PBS and Tofacitinib-treated samples (Fig. 6i). Both induction of CD25 by IL-15 and IL-7 as well as reduction of CD127 by IL-2, IL-7 and IL-15 were abrogated upon JAK inhibition, suggesting these cytokines modulated expression of their receptors directly.

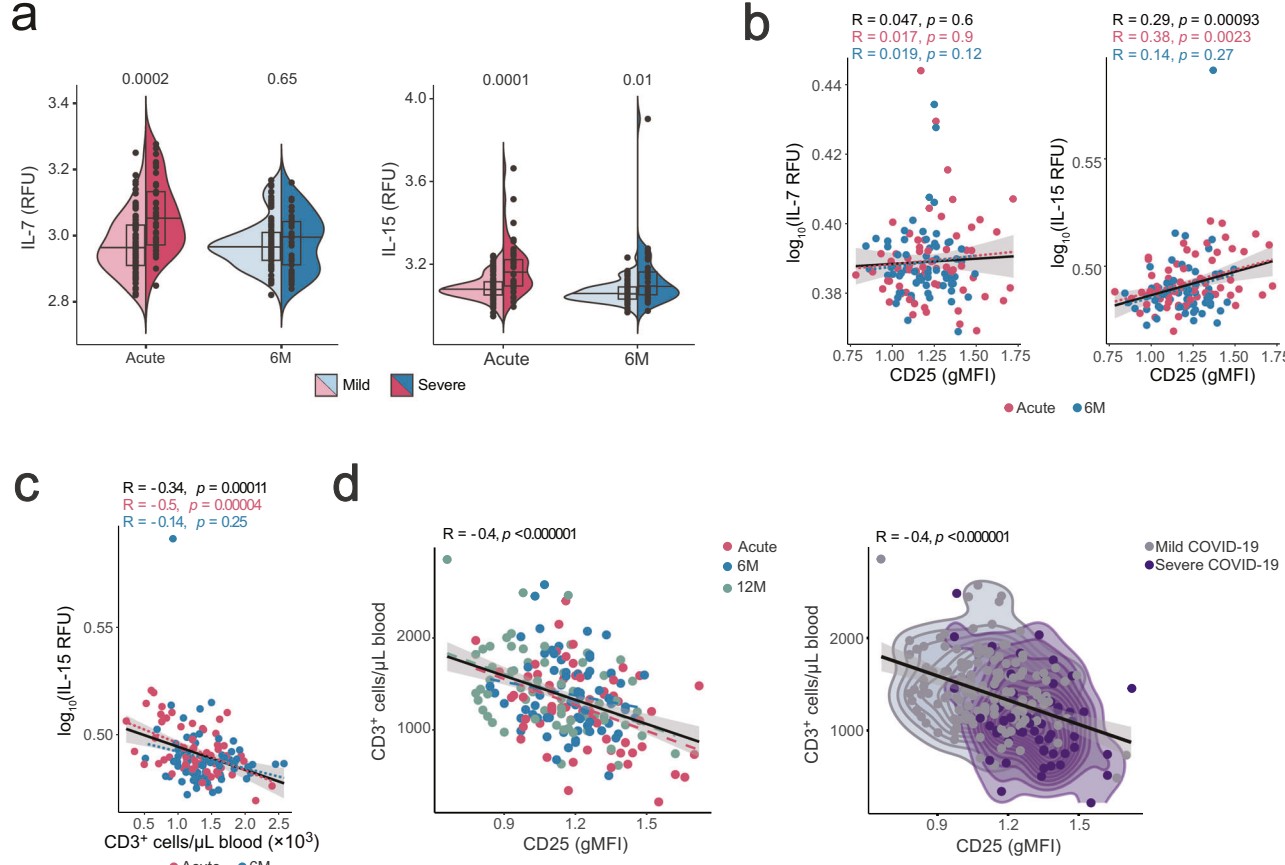

**Fig. 5 | Correlation of IL-2Rs and IL-7Rs on T cells upon viral infection with serum cytokines. a** Serum IL-7 and IL-15 levels in relative fluorescence units (RFU), measured by SomaScan platform using IL-7- and IL-15-specific aptamers in COVID-19 patients with mild (*n* = 76) and severe disease (*n* = 37) during acute infection and 6 M follow-up. Statistical analysis by two-tailed t-test. Boxplots are centered around the median; lower and upper hinges correspond to first and third quartiles, respectively; and lower and upper whiskers extend from hinge to smallest and largest values no further than 1.5× the IQR, respectively. **b** Correlation of CD25 expression on T cells with IL-7 or IL-15 concentrations in COVID-19 patient sera, depicted as overall trend (black line), as well as data points during acute infection (magenta; *n* = 64) and at 6-months follow-up (blue; *n* = 69). **c** Correlation of serum IL-15 levels and total T cell counts in blood of COVID-19 patients during acute infection (*n* = 64) and at 6 M follow-up (*n* = 69). **d** Correlation (*p* = 7.2 × 10⁻⁹) of CD25 expression on T cells and T cell blood counts of COVID-19 patients during acute infection (*n* = 64), and at 6 M (*n* = 69) and 12 M follow-up (*n* = 60). Correlation between variables is visualized by linear regression, with R indicating the two-sided Pearson correlation coefficient with corresponding *p* values. Gray band indicates 95% confidence interval. Source data are provided in the Source Data file.

## TCR activation facilitates IL-2-mediated CD25 upregulation

Since we observed a discrepancy of CD25 levels on SARS-CoV-2-specific CD8⁺ T cells compared to bystander-activated T cells following COVID-19 infection (Figs. 3f and 4), we assessed the role of TCR stimulation in aiding cytokine-mediated modulation of IL-2R and IL-7R. To this end, we preactivated enriched T cells by anti-CD3 and αCD28 antibodies (αCD3/αCD28) before subjecting them to cytokine stimulation (Fig. 7a and Supplementary Fig. 13a, b). Without preactivation, IL-7 and IL-15 robustly increased CD25 (Fig. 7a). IL-7 was more potent on CD4⁺ T cells and IL-15 induced the highest levels of CD25 on CD8⁺ T cells, reflecting their differential dose responses observed during short-term cytokine stimulation. Without preactivation, IL−2 did not result in significant CD25 upregulation in CD8⁺ T cells and only in subtle CD25 upregulation in CD4⁺ T cells. Interestingly, despite these differences in the capacity of inducing CD25, all cytokines led to similar degrees of proliferation irrespective of preactivation. IL-2 and IL-15 only minimally affected CD127, which was most reduced after IL-7 treatment (Fig. 7a). After αCD3/αCD28 preactivation, IL-2 significantly increased CD25 on both T cell subsets, albeit to a lesser degree than IL-15 and IL-7 (Fig. 7a).

Quantifying the fraction of different proliferating and CD25-expressing subsets over the course of stimulation revealed distinct phenotypic states depending on cytokine stimulus, T cell subset and TCR-mediated preactivation (Supplementary Fig. 13c). Without preactivation, CD25 upregulation following stimulation with IL-7 or IL-15, but not with IL-2, preceded Ki-67 expression on both CD4⁺ and CD8⁺ T cells. Interestingly, this population of Ki-67⁻ CD25⁺ was more frequent and persisted longer in CD4⁺ T cells compared to CD8⁺ T cells (Supplementary Fig. 13d) the latter of which transitioned faster to a Ki-67⁺CD25⁺ phenotype and gradually increased in frequency over time (Supplementary Fig. 13c). By contrast, IL-2 stimulation preferentially induced a CD25⁻ Ki-67⁺ population after 96 h of stimulation (Supplementary Fig. 13c). After αCD3/αCD28 preactivation, the fraction of Ki-67⁺ CD25⁺ was consistently higher compared to corresponding samples stimulated by cytokine only (Supplementary Fig. 13e). Furthermore, all cytokines tested, including NARA1leukin, further increased the proportion of the Ki-67⁺ CD25⁺ subset compared to cells that received preactivation but no cytokine signal (Supplementary Fig. 13c–f). Thus, preactivation increased the overall magnitude of CD25 induction by γc cytokines and was associated with prolonged elevation of CD25 after removal of cytokine stimulation (Fig. 7b).

In contrast to conventional T cell populations where IL-2 without preactivation only modulated CD127 but did not induce significant expression of CD25, Treg cells readily upregulated CD25 during IL-2 stimulation, even in the absence of TCR preactivation (Fig. 7c). To confirm these observations also in an in vivo setting, we characterized

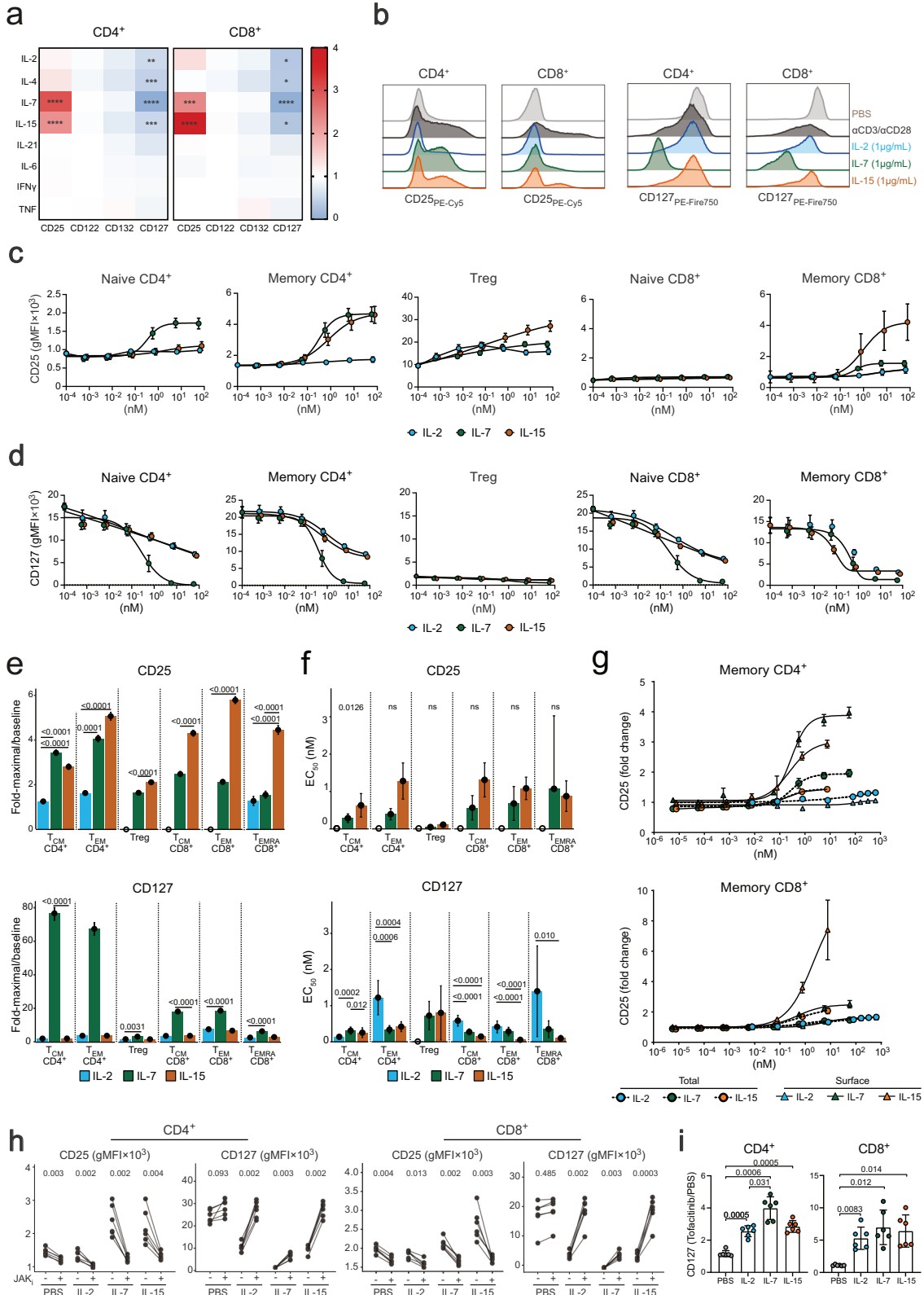

the IL-2R and IL-7R profiles of T cells from systemic lupus erythematosus (SLE) patients undergoing IL-2 immunotherapy. 12 female SLE patients received low-dose IL-2 injections on 5 consecutive days (Fig. 7d)[19]. Comparing CD25, CD127, and Ki-67 levels on different T cell subsets of these patients at baseline and after IL-2 treatment showed a similar pattern as in our in vitro data (Fig. 7e and Supplementary Fig. 13g, h). Whereas several subsets of both Treg and conventional

T cells showed increased levels of Ki-67 and a decrease of CD127, CD25 was exclusively upregulated on Treg cells. Collectively, these data indicate that CD25 expression on human conventional T cells is differentially regulated, with marked increase upon TCR signaling, modest upregulation upon IL-7 and IL-15 stimulation, but an inherent resistance to increase upon IL-2 signaling in the absence of concomitant TCR stimulation.

**Fig. 6 | Modulation of IL-2R and IL-7R expression on human T cells by IL-7 and IL-15 in vitro. a** Heatmap depicting fold change of indicated receptor subunits following continuous stimulation with indicated cytokines for 48 h in vitro over PBS-treated controls (*n* = 4). *P* values were determined by paired, two-sided Wilcoxon signed-rank test and adjusted for multiple comparisons using the Benjamini-Hochberg method. **b** Histograms of CD25 and CD127 on CD4⁺ and CD8⁺ T cells following activation with indicated stimuli. Modulation of CD25 (**c**) and CD127 (**d**) levels on indicated CD4⁺ and CD8⁺ T cell subsets after stimulation with indicated cytokines for 48 h in vitro (*n* = 15 healthy individuals). Symbols indicate median ± standard deviation (SD). Fold-maximal levels over baseline expression (**e**) and EC₅₀ values (**f**) of CD25 and CD127 after stimulation with titrated concentrations of IL-2, IL-7 and IL-15 for 48 h in vitro. Fold-maximal levels over baseline expression were calculated by the ratio of the upper and lower plateau of the dose-response curve for each cytokine according to a four-parameter log-logistic model (*n* = 15 healthy individuals). In **f**, open symbols represent data points where the model failed to reach convergence and no EC₅₀ value was determined. Maximal response values

and EC₅₀ values were compared using one-way ANOVA and *p* values were determined by Tukey's multiple comparison testing. Symbols indicate median ± standard deviation (SD). **g** CD25 surface expression (solid lines) and total CD25 levels (dashed lines) of memory CD4⁺ and CD8⁺ T cells after stimulation with indicated cytokines for 48 h in vitro (*n* = 3 healthy individuals). Symbols indicate median ± standard deviation (SD). **h** gMFI of CD25, CD122 and CD127 of CD4⁺ and CD8⁺ T cells after continuous stimulation with indicated cytokines (1 μg/mL) for 24 h in vitro, in the absence (−) or presence (+) of the Janus kinase inhibitor (JAKᵢ) Tofacitinib (1 μM), (*n* = 6 healthy individuals). **i** Fold-change difference in CD127 expression on T cells, treated with Tofacitinib over PBS, following stimulation with indicated cytokines as in (**h**) (*n* = 6 healthy individuals). For **g**, **h**, statistical testing was conducted using two-way ANOVA and *p* values were determined by Tukey's multiple comparison testing. *\*p* < 0.05; *\*\*p* < 0.01; *\*\*\*p* < 0.001; *\*\*\*\*p* < 0.0001. Dose response assays were done in three independent experiments. Source data are provided in the Source Data file.

## Discussion

In this study, we conducted a detailed analysis of CD25, CD122 and CD127 in peripheral lymphocytes at steady-state and following acute SARS-CoV-2 infection in humans. By investigation of antigen-specific cells during infection and after vaccination as well as in vitro and in vivo cytokine stimulation, we further delineated antigen-dependent and cytokine-driven mechanisms in the differential regulation of γc cytokine receptors. Our results provide several new insights into the distribution and modulation of IL-2Rs and IL-7Rs, which are relevant for immune regulation and cytokine immunotherapy.

Firstly, we provided a comprehensive quantification of CD25, CD122, CD127 and CD132 in 18 lymphocyte subsets at steady-state. Previous reports on IL-2R distribution on different peripheral lymphocytes in humans showed high levels of CD25 in Treg cells and high levels of CD122 in NK cells[7,40]. Our data further support and refine these observations, including the presence of CD25 on CD56^bright NK cells and the distribution of CD25 on CD4⁺, CD8⁺ T cell and certain B cell subsets. Furthermore, we found significant differences in CD122 abundance on peripheral T cell subsets, with the highest levels observed in CD8⁺ memory cells and lower levels in CD4⁺ T cells, as in mice[44,45]. To our knowledge, these data represent the most detailed characterization of IL-2R and IL-7R distribution at steady-state in human peripheral lymphocytes and reveal a finely graded distribution of CD25 and CD122 expression in T cell subsets. Even subtle changes in IL-2R expression are relevant for balancing immune responses, as recently suggested for a subset of naive CD4⁺ T cells with slightly increased CD25 expression in patients with multiple sclerosis[46]. As the IL-2R system harbors promising targets for immunostimulatory and immunosuppressive treatments[29,47–49], understanding the distribution of IL-2R subunits is key to tailor targeted therapies.

Secondly, following acute SARS-CoV-2 infection in humans, we observed protracted changes of CD25, CD122 and CD127 levels, persisting for more than 6 months into recovery. Experiments of acute viral infection of mice showed rapid and marked upregulation of CD25 and CD122 in virus-specific CD8⁺ T cells in secondary lymphoid organs, which was accompanied by a decrease of CD127[10,50]. However, these changes were present only for a few days after virus clearance. By contrast, we found more subtle yet long-lasting changes in the IL-2R and IL-7R systems after acute SARS-CoV-2 infection in humans. These changes were observed on both antigen-specific and bystander T cells, both in naive and memory T cells. While both naive and memory T cells can receive TCR signals via self-peptide/MHC interactions during steady-state[37,42], the herein described γc cytokine receptor alterations in both antigen-specific and bystander T cells are likely caused by a protracted increase in concentrations of IL-7 and IL-15 due to lymphopenia following acute viral infection. These findings suggest notable differences in the regulation of γc receptors in mice and humans. Moreover, we also found a prolonged increase in T cell

activation and proliferation markers on bystander T cells 6 months after SARS-CoV-2 infection. Concomitantly, we observed a pronounced reduction of naive CD4⁺ and CD8⁺ T cells 6 months after COVID-19, which further indicated shifts in the T cell compartment. These cellular changes were accompanied by a sustained increase in serum IL-7 and IL-15 levels in patients after severe COVID-19 up to 6 months after acute infection. We and others have previously shown profound perturbation of T cell homeostasis during acute COVID-19, including marked T cell lymphopenia, which was associated with increased serum IL-7 concentrations[39,51]. Although a quantitative and functional recovery of the peripheral T cell compartment was observed after recovery from COVID-19, a slight reduction of T cell counts persisted for several months[38], potentially leading to an increased availability of IL-7 and IL-15. Increased cytokine signaling following the acute infection stage may, in turn, contribute to the pathogenesis of post-acute infection pathologies, such as LC. In line, we found that SARS-CoV-2-specific CD8⁺ T cells of LC patients with persistent symptoms 12 months after acute disease showed higher CD25 levels compared to convalescent individuals. Notably, both SARS-CoV-2-specific and bystander T cells carried increased CD25 expression at 6 M follow-up, whereas at 12 M follow-up, elevated CD25 expression was observed only in SARS-CoV-2-specific CD8⁺ T cells in LC patients, which suggests either persistent lymphocyte dyshomeostasis and dysregulated cytokine signals or the involvement of TCR-mediated signals in LC. The latter could indicate persistent SARS-CoV-2 antigen, which has been detected in several human tissues[42,43]. Alternatively, the elevated CD25 levels at 12 M follow-up on SARS-CoV-2-specific CD8⁺ T cells could be explained by the more recent TCR activation of these cells, thus pre-activating them for cytokine-mediated stimulation. In line with this suggestion, our in vitro assays showed a more vigorous response to IL-7 and IL-15 in cells pre-activated by TCR signals, compared to those receiving cytokine signals only. We showed that these cytokines were sufficient to alter CD25 and CD127 expression in vitro, suggesting continuous and aberrant cytokine signaling, caused by lymphopenia-induced dysregulation of cytokine environment, as a mechanism for persistent receptor alterations following COVID-19. A similar HLA-DR^hi CD122^hi CD127^lo phenotype has previously been reported during T cell reconstitution after lymphocyte-depleting antibody treatment, where CD127 levels normalized only after 24 months[52]. Thus, profound T cell depletion during COVID-19 may similarly affect the T cell compartment for long periods, with increased IL-7 and IL-15 concentrations altering γc receptor levels. On the other hand, TCR engagement in vitro resulted in upregulation of CD25 and downregulation of CD127 in CD4⁺ and CD8⁺ T cells, as previously described[20]. In line with this, profiling cytokine receptors on antigen-specific T cells upon antigen rechallenge in vivo revealed a transient induction of CD25 and a more durable reduction in CD127 levels. As previously established in the context of acute lymphocytic

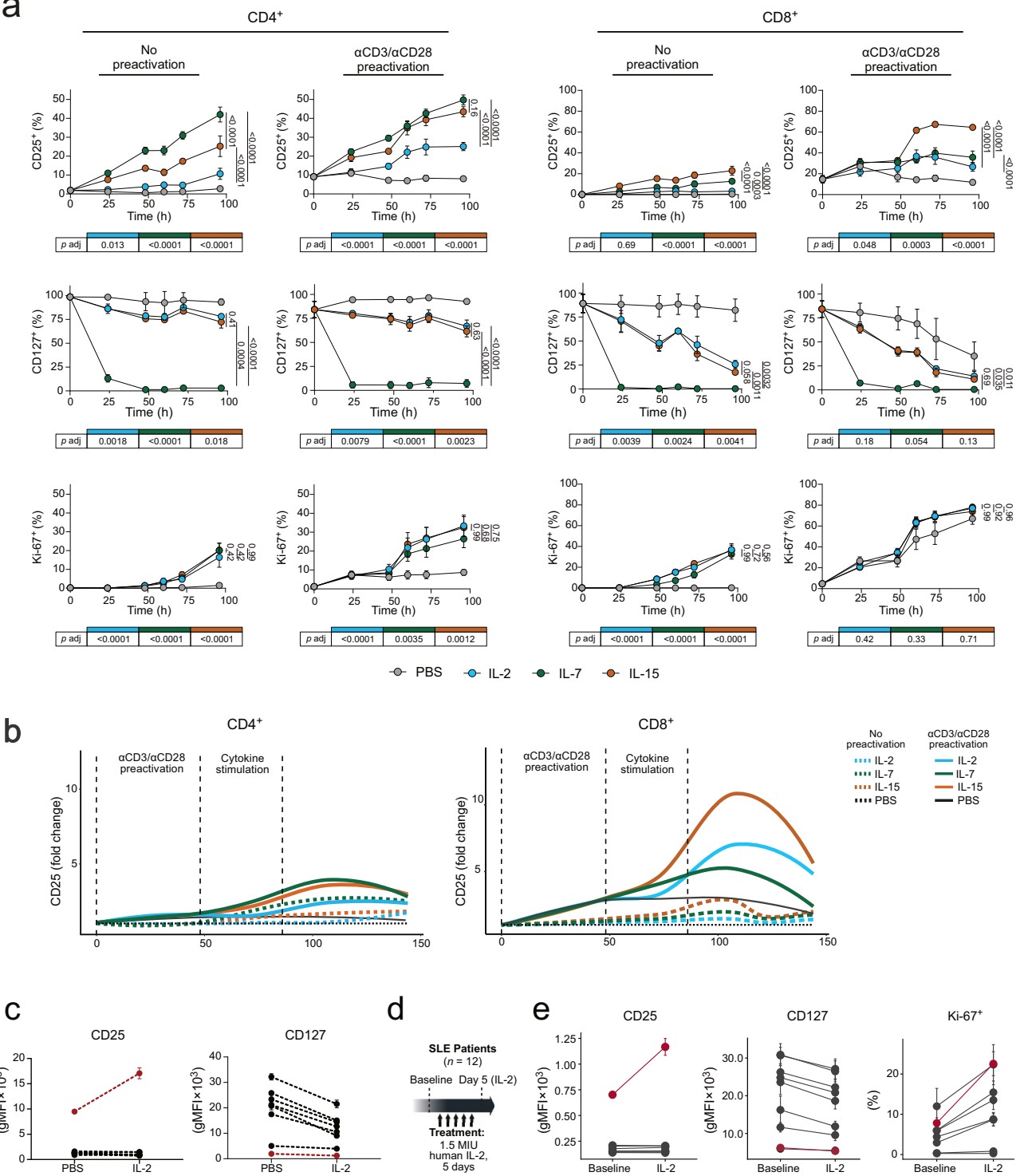

**Fig. 7 | CD25 levels on T cell subsets following long-term cytokine stimulation in vitro and in vivo. a** Fractions of CD25+, CD127+, and Ki-67+ CD4+ and CD8+ T cells during continuous stimulation with indicated cytokines, with or without prior activation by αCD3 and αCD28 antibodies for 48 h (*n* = 4 healthy individuals, two independent experiments). Frequencies at endpoint of stimulation were compared by 2-way ANOVA and *p* values were determined by Tukey's multiple comparison testing. Adjusted *p* values in the tables indicate comparison of cytokine stimulation to corresponding PBS control. **b** CD25 expression on CD4+ and CD8+ T cells after a pulse of cytokine stimulation for 36 h, following preactivation by αCD3 and αCD28 antibodies (*n* = 4 healthy individuals, two independent experiments). Data is shown

normalized to respective PBS control without preactivation. **c** CD127 and CD25 levels on T cells stimulated by IL-2 (0.65 nM) for 48 h in vitro (*n* = 15 healthy individuals, three independent experiments). Each data pair represents a distinct memory T cell subset. Treg cells are highlighted in red. **d** Schematic of SLE patient cohort receiving low-dose, 1.5 million international units (1.5 MIU) recombinant human IL-2, as previously described[19]. **e** Percentage of Ki-67+ cells and gMFIs of CD127 and CD25 on different T cell subsets of SLE patients (*n* = 12) before and after IL-2 immunotherapy. Each data pair represents a distinct memory T cell subset. Treg cells are highlighted in red. Symbols indicate median ± standard deviation (SD). Source data are provided in the Source Data file.

choriomeningitis virus (LCMV) infection in mice, CD25 expression on LCMV-specific CD8[+] T cells was associated with a short-lived effector-like phenotype[10]. Conversely, increased CD25 expression on LCMV-specific CD4[+] T cells positively correlated with increased counts of LCMV-specific memory T cells 30 d following immunization, in line with previous studies demonstrating additional IL-2 signaling checkpoints in establishing robust CD4[+] T cell memory and modulating secondary T cell responses[53,54].

Stimulation with IL-7 and IL-15 in vitro increased CD25 levels, which have previously been associated with increased phosphorylation of STAT5, which in turn mediates *IL2RA* gene transcription[55]. Interestingly, even high doses of IL-2 failed to upregulate CD25 in the absence of TCR engagement. Contrarily, IL-2 was as efficient as IL-7 and IL-15 in inducing T cell proliferation and downregulating CD127 expression. This was observed not only during in vitro stimulation of T cells but also following treatment of SLE patients with IL-2 in vivo. The fact that IL-2 cannot induce CD25 expression in the absence of antigen stimulation may reflect the strict transcriptional safeguarding of the *IL2RA* gene[56], as *IL2RA* transcription is tightly controlled to prevent excessive activation by autocrine and paracrine IL-2 signals. Several distinct superenhancer elements have been described, which modularly boost transcription and are engaged by different upstream signaling pathways[56,57]. Cell subsets with low homeostatic CD25 expression, such as memory T cells, which are largely devoid of the high-affinity trimeric IL-2R, failed to upregulate CD25 in response to IL-2. TCR preactivation upregulates CD25 and thus increases the formation of the high-affinity trimeric IL-2R, which subsequently enables *IL2RA* transcription. By contrast, IL-2 signaling through the low-affinity dimeric IL-2R appears to be insufficient to induce *IL2RA* transcription. Interestingly, however, IL-15 signaling through the dimeric IL-2R was able to upregulate CD25, illustrating a quantitative or qualitative difference of IL-2 and IL-15 signaling via the dimeric IL-2R.

Altogether, our findings provide a very detailed characterization of IL-2R and IL-7R distribution on human peripheral lymphocyte subsets at steady-state and following acute viral infection and cytokine stimulation. We present evidence of long-lasting changes of IL-2Rs and IL-7Rs after acute viral infection in humans and provide mechanistic insights into the modulation of these receptors. Considering the crucial roles of γc receptors in shaping adaptive immune responses, these results have important implications for delineating mechanisms of immune homeostasis and targeted cytokine immunotherapy. Furthermore, sustained alterations in these receptors leading to aberrant signaling in affected cells may result in low-grade systemic inflammation, as observed in certain post-acute infection syndromes.

## Methods

### Human subjects
Adult individuals were recruited following written informed consent for blood sampling, as approved by the Cantonal Ethics Committee of Zurich (BASEC #2016-01440)[31–38]. Individuals with negative history and serology for SARS-CoV-2 infection were included as healthy controls (Supplementary Table 1). Participants with reverse transcriptase quantitative polymerase chain reaction-confirmed SARS-CoV-2 infection were included during acute infection and followed up after approximately 6 months and one year (Supplementary Table 1). Maximum disease severity was determined according to World Health Organization classification criteria[58], with severe disease defined by the requirement for supplemental oxygen therapy. For the booster vaccination cohort, healthy volunteers were recruited who had previously received at least three doses of COVID-19 vaccination and did not report a SARS-CoV-2 infection in the months prior to the booster vaccine. Female systemic lupus erythematosus (SLE) patients were recruited following written consent. These patients had a confirmed diagnosis of SLE according to the American College of Rheumatology classification criteria[59], stable corticosteroid dose for at least 4 weeks

prior to enrollment, unchanged immunosuppressive medication for at least 4 weeks prior to enrollment and no major impairment in cardiac, pulmonary, renal, and hepatic organ functions.

### Routine laboratory testing
Routine laboratory testing was performed at accredited laboratories at University Hospital Zurich. Briefly, serum samples were obtained with Vacutainer CAT serum tubes (BD Biosciences). Semi-quantitative SARS-CoV-2 spike S1-specific IgA and IgG were measured with commercially available enzyme-linked immunosorbent assays (ELISA) (Euroimmun)[33]. IL-1β, IL-6, IFN-γ, and soluble CD25 (sIL-2Rα) were determined by ELISA (Quantikine, R&D Systems). Cytometric bead arrays were used for quantification of IL-10 (Flex Set, BD Biosciences) on a Navios Cytometer (Beckman Coulter) and TNF (R&D Systems) on a MagPix instrument (ThermoFisher). Absolute lymphocyte counts were obtained using Flow Set Pro Fluorospheres calibration beads (Beckman Coulter) on a Navios Cytometer (Beckman Coulter).

### Spectral flow cytometry
Venous blood was collected in EDTA vacutainer tubes (BD Biosciences), and peripheral blood mononuclear cells (PBMCs) were isolated by gradient centrifugation in Sepmate tubes (STEMCELL) with Lymphodex solution (Inno-Train Diagnostik). Five million PBMCs were stored in 1 ml fetal bovine serum (FBS) containing 10% dimethylsulfoxide (DMSO) in liquid nitrogen until further use. Samples were thawed in pre-warmed RPMI 1640 medium, washed with phosphate-buffered saline (PBS) and stained for viability with ZombieUV (Biolegend) in PBS containing Human TruStain FcX Fc-blocking reagent (Biolegend) for 30 min at 4 °C. After washing, chemokine receptors were stained with fluorochrome-labeled monoclonal antibodies (mAbs) for 20 min at 37 °C in PBS containing 1% FBS, 2 mM ethylenediaminetetraacetic acid (EDTA) and 10% brilliant stain buffer (BD Biosciences). After washing, the remaining surface markers were stained for 30 min at room temperature (RT) in PBS containing 1% FBS, 2 mM EDTA and 10% brilliant stain buffer (BD Biosciences). For measurements of intracellular markers, cells were incubated in a fixation permeabilization solution (eBioscience FoxP3/transcription factor staining buffer) for 60 min at RT, followed by washing and staining with mAbs for 30 min at RT.

For comparative staining of surface and total cytokine receptor levels after in vitro activation, each condition was split to two. Surface receptors were stained using fluorescently-labeled mAbs, followed by fixation with a fixation–permeabilization solution (eBioscience FoxP3/transcription factor staining buffer) for 60 min at RT, as described above to assure equal background autofluorescence. Total receptor levels were quantified by subjecting cells to fixation and permeabilization prior to staining with fluorescently labeled mAbs. A complete list of fluorochrome-labeled mAbs is available in Supplementary Table 2. After washing, samples were resuspended in PBS containing 1% FBS and 2 mM EDTA and acquired on a Cytek Aurora spectral flow cytometer using the SpectroFlo software.

### T cell stimulation assay
T cells were enriched from freshly isolated PBMCs by two-step magnetic-activated cell sorting (MACS), using negative selection with anti-CD14 microbeads in MACS LD columns, followed by positive selection with anti-CD4 and anti-CD8 microbeads in MACS LS columns, according to the manufacturer's instructions (all from Miltenyi Biotec). $1 \times 10^5$ T cells were incubated in 100 µl RPMI containing 10% FBS, penicillin/streptomycin, sodium pyruvate, L-glutamine, and MEM non-essential amino acids (all from Gibco) in 96-well U-bottom plates. For mAb stimulation with anti (α)-CD3 (clone OKT3, Biolegend) or αCD28 (clone CD28.2, Biolegend), plates were pre-coated overnight with 2 µg/ml mAb in 50 µl PBS at 4 °C. For cytokine stimulation, human recombinant IL-2 (Proleukin®, Clinigen Healthcare), IL-7 (Peprotech), and IL-15

(Peprotech) were used at the indicated concentrations. Certain stimulations were performed using a CD122-directed anti-human IL-2 mAb (clone NARA1, 50 μg/ml)[18]. For continuous stimulation, cells were incubated for the indicated time points at 37 °C in 5% $CO_2$. For pulsed stimulation, cells were incubated at 37 °C for 48 h or 72 h, transferred to fresh plates, washed twice with pre-warmed medium, and further cultured in full medium without stimulation. After incubation, samples were processed for flow cytometry analysis, as described above.

## MHC multimer staining

After human leukocyte antigen (HLA) typing, individuals carrying an HLA-A*02:01, HLA-A*01:01, HLA-A*03:01, HLA-A*24:02, HLA-B*07:07, HLA-DRB1*15:01, HLA-DRB1*04:01 or HLA-DRB1*07:01 allele were selected for assessment of virus-specific T cells following COVID-19 mRNA vaccination with BNT162b2 or acute SARS-CoV-2 infection. Samples were thawed in pre-warmed RPMI medium, washed and incubated in PBS containing Human TruStain FcX Fc-blocking reagent (Biolegend) for 10 min at 4 °C. After washing, cells were incubated with fluorochrome-labeled MHC dextramers loaded with different virus-derived peptides (Supplementary Table 3) in PBS containing 1% FBS and 2 mM EDTA for 20 min at RT. Thereafter, mAbs for surface marker staining were added, and samples were again incubated for 30 min at RT. After washing, intracellular markers were stained, and samples were processed, as described above.

## Serum processing and SomaScan proteomics

For serum isolations, venous blood was sampled using BD Vacutainer CAT serum tubes (Becton Dickinson, Franklin Lakes, NJ). Samples were centrifuged (1100 × g, 4 °C, 10 min) and stored at −80 °C until further use. Serum samples of healthy controls (n = 39) and paired samples of COVID-19 patients during acute infection (n = 113) and at 6-month follow-up were analyzed using the SomaScan platform (version 4)[60]. Measurements were carried out using 7335 modified single-stranded aptamers, including aptamers specific for 6596 unique human proteins and 46 internal controls[34,60,61]. All measurements passed manufacturer-defined quality control standards. Relative fluorescence units were $log_{10}$-transformed for analysis.

## Olink cytokine measurements

Serum samples were collected as described above and analyzed using the commercially available proximity extension assay-based platform 92-marker inflammation panel (Olink® Proteomics). Inflammation markers were included, which passed internal quality control and where more than half of the samples analyzed exceeded the detection limit.

## Computational batch normalization

Samples assessed for CD25, CD122 and CD127 quantification by flow cytometry were processed in nine separate batches. All samples of the same individual were analyzed in the same batch to facilitate intra-individual comparability. To allow for comparisons between individuals and to conduct correlation analyses with other measurements, computational batch normalization was performed. To this end, an identical control sample was thawed, stained and acquired with every batch. After per batch compensation and pre-gating in FlowJo software (version 10.8.0), FCS files were processed in R Studio (2022.02.1) running R (4.1.3) using an established batch normalization pipeline for flow cytometry data[62,63]. Briefly, arcsinh transformation was applied using *FlowVS* package (1.26.0) with automatically calculated and manually adjusted cofactors. Transformed control files were clustered using lineage markers CD56, CD3, CD4, CD8, CD45RA and CCR7 in 5 × 5-dimension FlowSOM clustering with seven metaclusters. Cell frequencies of metaclusters were not significantly affected by batch effect[62]. Using *CytoNorm* package (0.0.7) a normalization model was calculated from control samples, which was subsequently applied on

study samples. Downstream analysis of normalized FCS files was performed in FlowJo software using identical gating for all samples.

## Statistics

Flow cytometry data were analyzed in FlowJo software (10.8.0). All statistical analyses were performed in R Studio (2023.06.0) running R (4.4.1). Between-group comparison of independent groups was obtained with Mann-Whitney U tests, and paired testing was performed using Wilcoxon signed-rank tests. *P* values were adjusted for multiple comparisons using the Benjamini-Hochberg method unless otherwise specified. Spearman's rank correlation was used to assess associations of numeric variables. Data was visualized with *ggplot2* (3.3.5), *corrplot* (0.90), and *ggpubr* (0.4.0). In violin plots, medians are shown as horizontal lines. Simple linear regression models were used for visualization of regression lines.

## Reporting summary

Further information on research design is available in the Nature Portfolio Reporting Summary linked to this article.

## Data availability

All data needed to evaluate the conclusions in the paper are present in the paper or the supplementary materials. The raw flow cytometry data are protected and are not available due to data privacy laws. The processed data are available and deposited in the Zenodo repository https://doi.org/10.5281/zenodo.17377036. Source data are provided with this paper.

## Code availability

The code generated during the current study is available at https://github.com/BoymanLab/IL2R_IL7R_COVID.

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

## Acknowledgements

We thank the diagnostic laboratories of the University Hospital Zurich, Jakob Nilsson, Sara Cervia-Hasler, Lars C. Huber, Melina Stüssi-Helbling, Alain Rudiger, Esther Bächli, Alessandra Guaita, Claudia Meloni, Jennifer Jörger, Claudia Bachmann and Jana Epprecht for their support, and the members of the Boyman Laboratory for helpful discussions. The study overview graphics were created with BioRender.com. This work was funded by Swiss National Science Foundation grants 310030-172978, 310030-200669, 310030-212240 and 4078P0-198431 (to O.B.) and NRP78 Implementation Programme (to C.C. and O.B.), Digitalization Initiative of the Zurich Higher Education Institutions Rapid-Action Call #2021.1_RAC_ID_34 (to C.C.), Swiss Academy of Medical Sciences grants 323530-191230 (to Y.Z.), 323530-191220 (to C.C.), and 323530-177975 (to S.A.), Forschungskredit Candoc of University of Zurich FK-20-022 (to S.A.), Young Talents in Clinical Research Project Grant (YTCR 08/20) by the Swiss Academy of Medical Sciences and Bangerter Foundation (to M.E.R.), the Clinical Research Priority Program of University of Zurich for CRPP CYTIMM-Z (to O.B.), the Pandemic Fund of University of Zurich (to O.B.), and an Innovation Grant of University Hospital Zurich (to O.B.).

## Author contributions

L.C. contributed to patient recruitment, performed experiments, and analyzed and interpreted data. P.T. contributed to patient recruitment, performed experiments, and analyzed and interpreted data. A.I. performed experiments. Y.Z. contributed to patient recruitment and analyzed data. S.A. performed experiments and analyzed data. C.C.H. and M.E.R. contributed to patient recruitment and clinical management. O.B. conceived the project and interpreted the data. L.C. P.T., and O.B. wrote the manuscript. All authors edited and approved the final draft of the article.

## Competing interests

O.B. is a shareholder of Anaveon AG, which develops IL-2 immunotherapies for cancer. O.B. and M.E.R. hold patents on improved IL-2 formulations and are shareholders of Seito Biologics AG, which develops improved IL-2 immunotherapies for autoimmune diseases. The other authors declare no competing financial interests.
