## [Transparent Peer Review file · Nature Communications]

Dysregulation of homeostatic cytokine receptors drives prolonged T cell activation following acute viral infection in humans

Corresponding Author: Professor Onur Boyman

Version 0:

Reviewer comments:

Reviewer #1

(Remarks to the Author)

The manuscript "Dysregulation of homeostatic cytokine receptors drives prolonged T cell activation following acute viral infection in humans" by Ceglarek et al. nicely demonstrates how common gamma chain cytokine (IL-2, IL-7, and IL-15) and their receptors are alternately utilized/ regulated by lymphocyte (particularly T cells) in the context of viral infection. The study makes use of peripheral blood lymphocytes from both healthy controls and SARS-CoV-2 vaccination/ infection cohorts. Their principle findings are that T cells alter their expression of common gamma chain cytokine receptors in response to infection. This includes common gamma chain cytokines altering the expression of other common gamma chain cytokine receptors. While many of these findings may be variably established in mouse models it is relevant to establish them and understand their interactions in humans. Overall, the manuscript and data presented are sound. I am recommending to accept the manuscript following revision for one critical point and a few minor (discretionary) points.

Critical Revision:

My primary hangup with this manuscript is the interrogation and discussion of "bystander" responding cells. Certainly bystander responses are interesting and relevant to these sorts of interrogations but the authors have insufficiently delineated how bystander cells are responding. By using only a single dextramer the authors miss out on all other antigen specific (non-bystander) responses. Consequently their dextramer negative cells are not bystander but rather an amalgamation of both infection specific and non-specific responses (including both naive and memory). This prevents them from delineating whether the bystander cells are truly behaving in the manner described. If the authors want to pursue the bystander angle they should perform analyses on dextramer+ cells that are specific to non-SARS-CoV-2 pathogens. Ideally this would be toward something like influenza or RSV where the subject is unlikely to have active co-infection and not persistent infections (e.g., CMV or EBV) which may exit latency during high stress (i.e., a secondary[SARS-CoV-2] infection). Alternately, the authors can separate out the naive and memory cells in the dextramer- population and rework their discussion of their present results to more accurately describe the composition of the dextramer negative cells and thoroughly discuss how this composition influences interpretation of their results.

Minor Revisions:

- 1) The title doesn't feel particularly accurate for the manuscript. Rather than dysregulation of cytokine receptors driving prolonged activation, I feel the authors data more accurately demonstrate that infection induced changes in homeostatic cytokine abundance alter expression of homeostatic cytokine receptors by T cells.
- 2) The acronym for SARS-CoV-2 (line 127) appears before definition (line 129)
- 3) It may be worth discussing that as pointed out (Fig 2H, SFig 2F) CD25 expression by dextramer+ CD4 T cells at D3 post vaccination correlated with the size of the CD4 T cell response at D30 but CD25 expression by dextramer+ CD8 T cells does not correlate with the size of the response.
- 4) The authors see no association with comorbidities between healthy and infected individuals. However, they are likely underpowered to identify relationships with specific comorbidities. They should instead consider using an aggregate comorbidity index rather than splitting individual comorbidities.
- 5) It is interesting that the naive T cells (which are presumably uninvolved in the infection are also changing their cytokine receptor expression (Fig S4). This may be related to the ongoing inflammatory milieu (inline with manuscripts narrative) and could be worth discussing.

- 6) The authors address how T cell movement into tissue may influence any numeric changes they observe while sampling the blood
- 7) In cytokine pulse experiments (Fig S7) does prolonging cytokine exposure prolong the rate of decay for the changes in receptor expression?
- 8) Figure 1 receptor expression is compared across cell types but it may also be useful to depict which cell populations have expression that is significantly increased over background.
- 9) Use of contours in Fig 2D,E make it difficult to readily interpret.
- 10) It may be worth commenting on how Tofacitinib seems to have little to no effect on IL-7R in the context of IL-7 stimulation.
- 11) It would be easier to interpret Fig 6A if the scales between the no activation and the CD3/CD28 stim were the same.
- 12) It would be informative to indicate which groups significantly deviate from the healthy (dashed line) cohort in Fig S3C.
- 13) Fig S9C may be more readily understood if rather than linking each stimulation condition at a given timepoint the authors display the results as a time course for each stimulation condition.

Summarily I like the manuscript and look forward to its eventual publication and hopefully the authors don't find these recommendations burdensome.

Reviewer #2

(Remarks to the Author)

In the manuscript, Ceglarek et al. performed a comprehensive analysis of gc cytokine receptor expression on each lymphocyte subset in healthy humans. Moreover, the authors show that CD25 expression levels are upregulated and remain high on T cells whereas CD127 levels downregulated and remain low for 6 months post SARS-CoV-2 infection and that these changes are associated with long-lasting alterations of IL-15 and to a lesser extent IL-7 levels in serum. Furthermore, they provide evidence showing that these homeostatic cytokines can directly upregulate and downregulate CD25 and CD127 expression on T cells, respectively, *in vitro*.

The manuscript is well written and can be an important addition to this research field. To consolidate their findings, I have the following comments and suggestions.

[Major]

(1) What are the functional consequences of prolonged CD25 upregulation and CD127 downregulation in human T lymphocytes? Can the authors add data showing the relationship between these changes and immunopathology in COVID-19 or other contexts?

(2) Memory T cells are generally defined as those that can survive for long in the absence of cognate antigen recognition, and indeed in mice, LCMV Armstrong-specific memory CD4 and CD8 T lymphocytes can persist long after antigen deprivation. On the other hand, it has been reported that SARS-CoV-2 viral RNA can persist for 2 years in humans. Based on these observations, do the authors think that they are looking at "authentic" memory cells in 6/12-month follow-up samples of the present study? Or do dextramer(+) T cells on these time points represent persistently antigen-stimulated "effector" cells? This discussion should be included in the manuscript.

(3) Fig 2D - G: Taking it into account that CoV2-Dex(+) T cells have the same or similar TCR specificity, what does the small "Ki67+ CD38+" subpopulation that accounts for CD25 upregulation on day 3 represent?

(4) Fig 3F and G: The authors seem to consider CoV2-Dex(-) and CoV2-Dex(+) populations as bystander-activated and cognate antigen-stimulated cells, respectively (e.g., lines 212, 303, 370, etc.), although not explicitly stated. This is inaccurate or at least confusing because CoV2-Dex(-) T cells may contain cells that are specific for other SARS-CoV-2 antigens than those listed in Table S3. Interpretation on CoV2-Dex(-) T cells should be refined throughout the manuscript.

(5) Fig 5D: The authors demonstrate that IL-7 strongly downregulates CD127 levels on T cells. Can they rule out the possibility that the binding of IL-7 to CD127 inhibited that of anti-CD127 mAb to CD127? Related to this point, Tofacitinib only minimally inhibited IL-7-induced CD127 downregulation whereas IL-2/15-triggered CD127 decrement was almost completely blocked by the same treatment (Fig 5H), suggesting that apparent CD127 downregulation in the presence of IL-7 might not be attributed to the effect of the same cytokine.

(6) Fig 5G: How did the authors analyze total (i.e., surface and intracellular) CD25 expression levels?

[Minor]

(7) The subheading "TCR signals without systemic inflammation" (line 124) is overstated. What they performed in this section is kinetic analysis of gc cytokine receptor expression in SARS-CoV-2 booster vaccination.

(8) Although the authors state "Following immunization CD25 induction precedes T cell activation markers and proliferation" (line 143), this description is incorrect because CD25 induction coincides with rather than precedes T cell proliferation on day 3 (Fig 2).

(9) What data do the authors refer to by "Paired analysis of CD25 levels in COVID-19 patients" (line 180)?

(10) The subheading "IL-7 and IL-15 mediate sustained changes of IL-2R and IL-7R in the absence of TCR activation" (line 231) is overstated because at least in this section, the authors just showed correlation but not causal association among IL-7/15 concentrations, CD25 levels, and T cell counts. Also, they cannot conclude "in the absence of TCR activation" in this section.

(11) The sample size (n) is not specified in the legend of Fig 6C.

Version 1:

Reviewer comments:

Reviewer #1

(Remarks to the Author)

The authors have convincingly provided data and rationale to address my previous comments so I am happy to recommend the article for acceptance.

minor note: I think that the data callout in line 254 is meant to be (Supplementary Fig. 6b) rather than (Supplementary Fig. 6e) though I suspect this can be fixed during the editorial stage prior to publication.

Reviewer #2

(Remarks to the Author)

This reviewer thanks the authors for making substantial efforts on this revision. I have two minor comments as follows. Otherwise all my concerns have been comprehensively and satisfactorily addressed, and I recommend acceptance of this important paper for publication.

(1) Related to my previous comment #5: While it is now clear that IL-7 can downregulate CD127 via JAK (Fig. 6i), it seems to be also true that IL-7-induced decrement of the CD127 MFI in Fig. 6d is at least in part due to obstruction of the relevant epitope by the same cytokine. I suggest to include a brief description of this issue in the main text.

(2) Related to my previous comment #9: In Fig. 3c, I suggest adding lines connecting data at three time points for each sample (as the authors did in Fig. S4e and S5) for readability of the sentence in line 186 (Paired analysis of CD25 levels...).

“Dysregulation of homeostatic cytokine receptors drives prolonged T cell activation following acute viral infection in humans” by Laura Ceglarek, Patrick Taeschler, Alp Inci, Yves Zurbuchen, Sarah Adamo, Carlo Cervia-Hasler, Miro E. Raeber and Onur Boyman

We thank the editors and reviewers for their assessment of our manuscript. We appreciate the suggestions, and we thank the editors for giving us the opportunity of submitting a revised version of our manuscript. We have now revised our manuscript according to the reviewers' comments and the editor's suggestions. The changes are highlighted in yellow in the manuscript file.

Please note the following nomenclature used for referring to the figures:

Fig. 1 refers to main Figure 1 of the manuscript.

Supplementary Fig. 1 refers to Supplementary Figure 1 of the manuscript.

Fig. P1 refers to Figure 1 of these point-by-point responses.

Reviewer 1

Summary:

The manuscript "Dysregulation of homeostatic cytokine receptors drives prolonged T cell activation following acute viral infection in humans" by Ceglarek et al. nicely demonstrates how common gamma chain cytokine (IL-2, IL-7, and IL-15) and their receptors are alternately utilized/ regulated by lymphocyte (particularly T cells) in the context of viral infection. The study makes use of peripheral blood lymphocytes from both healthy controls and SARS-CoV-2 vaccination/ infection cohorts. Their principle findings are that T cells alter their expression of common gamma chain cytokine receptors in response to infection. This includes common gamma chain cytokines altering the expression of other common gamma chain cytokine receptors. While many of these findings may be variably established in mouse models it is relevant to establish them and understand their interactions in humans. Overall, the manuscript and data presented are sound. I am recommending to accept the manuscript following revision for one critical point and a few minor (discretionary) points.

Response: We thank the reviewer for their generous and helpful comments. We have now addressed the reviewer's suggestions and comments in our revised manuscript as well as in our point-by-point responses to their specific points, as outlined below.

Specific Points:

1) My primary hangup with this manuscript is the interrogation and discussion of "bystander" responding cells. Certainly bystander responses are interesting and relevant to these sorts of interrogations but the authors have insufficiently delineated how bystander cells are responding. By using only a single dextramer the authors miss out on all other antigen specific (non-bystander) responses. Consequently their dextramer negative cells are not bystander but rather an amalgamation of both infection specific and non-specific responses (including both naive and memory). This prevents them from delineating whether the bystander cells are truly behaving in the manner described. If the authors want to pursue the bystander angle they should perform analyses on dextramer+ cells that are specific to non-SARS-CoV-2 pathogens. Ideally

this would be toward something like influenza or RSV where the subject is unlikely to have active co-infection and not persistent infections (e.g., CMV or EBV) which may exit latency during high stress (i.e., a secondary [SARS-CoV-2] infection). Alternately, the authors can separate out the naive and memory cells in the dextramer- population and rework their discussion of their present results to more accurately describe the composition of the dextramer negative cells and thoroughly discuss how this composition influences interpretation of their results.

Response: We thank the reviewer for this important consideration. Initially we reasoned that the frequency of SARS-CoV-2-reactive T cells within the SARS-CoV-2-dextramer (CoV2-Dex)-negative CD8⁺ T cell compartment is unlikely to be sufficiently large to significantly skew the expression levels of IL-2R and IL-7R in the entire polyclonal T cell population. Particularly in the memory phase, the median frequencies of SARS-CoV-2-reactive CD8⁺ T cells were estimated to range between 0.2% and 0.5%, based on different *in vitro* peptide pool stimulation assays (Cohen et al. *Cell Rep. Med* 2 2021; Dan et al. *Science* 371, 2021). Nevertheless, we do agree with the reviewer that with the data presented in Fig. 2F, G, we cannot formally exclude that receptor expression levels on CoV2-Dex⁻ cells are truly caused by T cell receptor (TCR)-independent cytokine signals and are not driven by SARS-CoV-2-specific T cells reactive to epitopes not covered by our CoV2-dextramer reagents. We have now addressed this in two ways, as outlined below.

Firstly, we sought to experimentally address the issue of bystander T cell activation. To this end, we performed another longitudinal profiling experiment of common gamma chain (γ c; also termed CD132) receptor subunits on a subset of our infection cohort (**Fig. P1; corresponding to new Fig. 4**). Following the reviewer's suggestion, we included different dextramers carrying different influenza A virus (IAV)- and Epstein-Barr virus (EBV)-derived peptides to identify IAV- and EBV-specific bystander T cells and investigate whether signs of EBV reactivation can be detected in the post-acute infection stage (**Fig. P1a, b**). CoV-2-specific CD8⁺ T cells showed CD25 and CD122 upregulation and CD127 downregulation. We observed the same trends, but in attenuated fashion, on both IAV- and EBV-specific CD8⁺ T cells (**Fig. P1c, d**). Thus, both IAV- and EBV-specific CD8⁺ T cells showed signs of activation, marked by activation-associated molecules, such as CD38, HLA-DR, granzyme B (GzB) and Ki-67 (**Figs. P1e and P2**). As expected, the most prominent shift toward highly activated clusters was observed in SARS-CoV-2-specific T cells during acute COVID-19 (**Figs. P1e and P2b**). Likewise, EBV- and IAV-specific CD8⁺ T cells adopted an activated phenotype, which was associated with high expression of CD122, a moderate increase in CD25 and downregulation of CD127 (**Fig. P1f**). Since EBV-specific cells could receive antigenic signals by latent viral antigens, we focused particularly on IAV-specific cells. On IAV-specific CD8⁺ T cells, upregulation of CD122 and downregulation of CD127 significantly correlated with the cells' activation status across all sampling timepoints, suggesting that aberrant expression of these homeostatic receptors correlated with prolonged bystander T cell activation (**Fig. P1g-i**). We have now included these data in the revised manuscript as Fig. 4 and adapted the corresponding results and discussion section accordingly (lines 243-266).

restriction prevented use of certain dextramers in all subjects. **e** UMAP projection of CoV2-, IAV- and EBV-specific CD8⁺ T cells in healthy individuals and COVID-19 patients, clustered based on activation markers or grouped by indicated virus specificities. **f** Heatmap depicting mean expression of indicated receptor subunits in different clusters of CoV2-, EBV- and IAV-specific CD8⁺ T cells. **g** Correlation between indicated receptor subunits and markers on IAV-specific CD8⁺ T cells. **h** Correlation of HLA-DR and CD122 and CD127 expression on IAV-specific CD8⁺ T cells, depicted as overall trend (black line), as well as data points during acute infection (magenta; $n = 17$), and at 6M (blue; $n = 17$) and 12M follow-up (green; $n = 17$). **i** HLA-DR, T-bet and Ki-67 expression on IAV-specific CD8⁺ T cells in COVID-19 patients ($n = 17$) at indicated timepoints.

Secondly, as suggested by the reviewer and in addition to looking at the total polyclonal and CoV2-Dex⁺ T cell subsets, we split the CoV2-Dex⁻ population into naive and memory T cell subsets (Fig. P3). Although there were minor differences between these polyclonal and defined antigen-specific T cell subsets, both naive and most memory subsets within total and CoV2-Dex⁻ T cells showed slightly changed CD25, CD127 Ki-67 proliferation marker expression at 6-month (6M) follow-up compared to the reference timepoints (Fig. P3a, c).

and 12M follow-up ($n = 22$). **c** Expression of CD25, CD127 and Ki-67 on naive and memory CoV-2–Dex– CD8⁺ T cells during acute COVID-19 ($n = 21$), and at 6M ($n = 21$) and 12M follow-up ($n = 21$).

2) The title doesn't feel particularly accurate for the manuscript. Rather than dysregulation of cytokine receptors driving prolonged activation, I feel the authors data more accurately demonstrate that infection induced changes in homeostatic cytokine abundance alter expression of homeostatic cytokine receptors by T cells.

Response: We appreciate the reviewer's suggestion. We have revisited our title and considered alternative titles. We agree with the reviewer that our data "demonstrate that infection induced changes in homeostatic cytokine abundance alter expression of homeostatic cytokine receptors by T cells". However, our data also demonstrate that T cells adopt an activated state for up to 12 months after infection, as demonstrated in Figs. 3–5, and Supplementary Figs. 5 and 6. After much consideration and evaluation of alternative titles, we think that our current title most accurately and succinctly captures these two aspects. Thus, we would be much obliged if the reviewer agreed to leaving our title as is.

3) The acronym for SARS-CoV-2 (line 127) appears before definition (line 129).

Response: We thank the reviewer for picking up on this. We have now corrected this error in our revised manuscript.

4) It may be worth discussing that as pointed out (Fig 2H, SFig 2F) CD25 expression by dextramer+ CD4 T cells at D3 post vaccination correlated with the size of the CD4 T cell response at D30 but CD25 expression by dextramer+ CD8 T cells does not correlate with the size of the response.

Response: It is certainly intriguing that CD25 levels on CD4⁺ T cells, but not on CD8⁺ T cells, correlated with the size of the T cell memory compartment. Following the reviewer's suggestion, we have now expanded on this point in the discussion of our revised manuscript (lines 474-480), also citing previous relevant publications in mice. These latter include the finding that IL-2 signals during priming and expansion resulted in increased contraction, whereas IL-2 administration during contraction decreased apoptosis and increased maintenance of virus-specific CD8⁺ T cells (Blattman, J. N. et al. *Nat Med* **9**, 2003). Furthermore, CD25-deficient CD8⁺ T cells were impaired in their secondary recall responses (Williams, M. A. et al. *Nature* **441**, 2006), CD25^{hi} virus-specific CD8⁺ T cells exhibited a more differentiated and short-lived effector-like phenotype compared to their CD25^{int} counterparts (Kalia, V. et al. *Immunity* **32**, 2010), and somewhat connecting these previous findings, autocrine IL-2 signals by virus-specific CD8⁺ T cells facilitated their secondary recall responses (Feau, S. et al. *Nat Immunol* **12**, 2011). In line with these data in mice, our data in human T cells demonstrate that the CD25^{hi} fraction (i.e. the cells with the highest 20% of CD25 expression) within CoV2–Dex⁺ CD8⁺ T cells 3 d post-vaccination showed increased expression of T-bet and GzB (**Fig. P4a, b**), and might thus represent short-lived effector cells. By contrast, a higher upregulation of CD25 on CD4⁺ T cells at 3 d post-vaccination was associated with a larger SARS-CoV-2-specific T cell compartment at 30 d post-vaccination (**Fig. 2h**). In line with previous reports (Feau, S. et al. *Nat Immunol* **12**, 2011; and McKinstry, K. K. et al. *Nat Commun* **5**, 2014), CD25 upregulation can contribute to the formation of trimeric IL-2Rs and facilitate additional IL-2 signals, thus, providing the cells with proliferative and anti-apoptotic queues.

5) The authors see no association with comorbidities between healthy and infected individuals. However, they are likely underpowered to identify relationships with specific comorbidities. They should instead consider using an aggregate comorbidity index rather than splitting individual comorbidities.

Response: We thank the reviewer for this suggestion and calculated a comorbidity index for each patient based on the Charlson Comorbidity Index (Charlson, M. E. et al. *Journal of Chronic Diseases* **40**, 1987). Subsequently, we aggregated these weighted indices for the respective groups, repeated the statistical testing and revised **Supplementary Table 1**. However, this analysis did not identify any significant associations with comorbidities.

6) It is interesting that the naive T cells (which are presumably uninvolved in the infection) are also changing their cytokine receptor expression (Fig S4). This may be related to the ongoing inflammatory milieu (in line with manuscripts narrative) and could be worth discussing.

Response: We thank the reviewer for this point and have now placed more emphasis on this point in the discussion of our revised manuscript (lines 201-203 and 437-443).

7) The authors address how T cell movement into tissue may influence any numeric changes they observe while sampling the blood.

Response: We thank the reviewer for pointing out this relevant consideration when analyzing human blood samples without paired tissue samples. Indeed, during acute infection, a significant fraction of activated T cells is located within tissues and inaccessible to analysis. However, we believe that the reduction of T cell counts and concomitant increase in serum IL-7 and IL-15 (as illustrated in **Fig. 5**) is primarily a sign of apoptosis-mediated lymphopenia rather than cell migration. In line with this hypothesis, we have previously described increased apoptosis, as indicated by increased percentages of cleaved-PARP and/or cleaved caspase 3, in both CD4⁺ and CD8⁺ T cells of the same patient cohort (Adamo et al. *Allergy* **76**, 2021). Furthermore, association of reduced T cell counts, CD25 expression levels and cytokines was still observed at 6M follow-up. Subtly reduced T cell counts, particularly in individuals who underwent severe COVID-19, 6 months after acute infection is unlikely to be caused by differential migration dynamics but rather reflects T cell lymphopenia (**Fig. P5d**). We have included these data and expanded on this point in the relevant result section of our revised manuscript (lines 302-306).

8) In cytokine pulse experiments (Fig S7) does prolonging cytokine exposure prolong the rate of decay for the changes in receptor expression?

Response: It is an interesting point to consider whether the duration of cytokine signals leads to a quantitative or qualitative difference in cytokine receptors and whether the decay rate in returning to basal expression is linear. To address the reviewer's point, we have now performed another set of *in vitro* stimulation experiments, including an additional group stimulated with a cytokine pulse for 48 h (**Fig. P6a–c**). As expected, initial levels of CD25 were slightly higher after 48 h compared to 24 h and continued to rise in the continuously stimulated samples (**Fig. P6c**). Removal of cytokines lead to progressive decrease of CD25 that reached basal levels around 167 h post-stimulation. To assess whether the decay of CD25 after a cytokine pulse of 24 h and 48 h was comparable, we assessed the difference of CD25 upregulation compared to unstimulated cells at each timepoint following cytokine removal, normalized to CD25 induction at cytokine removal. This allowed us to adjust for the initially higher levels of CD25 at 48 h compared to 24 h of stimulation. Although not reaching statistical significance, there was a tendency of higher lingering CD25 expression on memory T cells stimulated for 48 h compared to those receiving cytokine only for 24 h (**Fig. P6d**).

9) Figure 1 receptor expression is compared across cell types but it may also be useful to depict which cell populations have expression that is significantly increased over background.

Response: We thank the reviewer for this comment and agree that statistical comparison to the matched isotype controls will aid the interpretation of receptor expression levels depicted in **Fig. 1c–f**. To this end, we have now calculated and statistically assessed the mean log₂-fold change of signals for each receptor subunit on each cell subset in relation to their matched isotype control (**Fig. P7a, b**). For CD25, which is the receptor subunit with the lowest steady-state expression levels, we also included a paired comparison of fully stained samples and controls to appreciate even subtle expression levels of this receptor subunit on individual cell subsets. We have included these data in the revised manuscript (**Supplementary Fig. 1b, c**).

10) Use of contours in Fig 2D,E make it difficult to readily interpret.

Response: To make Fig. 2d, e better readable, we have now visually highlighted the relevant regions in the UMAPs by circling the relevant region of interest (**Fig. P8, corresponding to Fig. 2d, e**).

11) It may be worth commenting on how Tofacitinib seems to have little to no effect on IL-7R in the context of IL-7 stimulation.

Response: We thank the reviewer for pointing this out. Previously we displayed linear geometric mean fluorescence intensities (gMFIs) of CD127 expression following treatment with the Janus kinase (JAK) inhibitor Tofacitinib and cytokine stimulation. Following IL-7 stimulation, both PBS and JAK inhibitor-treated samples showed low CD127 levels. These data could be due to rapid internalization and downregulation upon IL-7 signaling. Or the data could be the result of a technical issue due to partial obstruction of the relevant epitope by recombinant IL-7. To address this latter point, we preincubated peripheral blood mononuclear cells (PBMCs) with IL-7 on ice for 20 min prior to staining with the surface staining mix, which also contained IL-7. Indeed, the presence of IL-7 decreased the signals of two different anti-CD127 antibody clones, namely hIL-7R-M21 and A019D5 (Fig. P9a, b). However, the residual CD127 staining signals were still significantly higher than the corresponding isotype controls. Furthermore, cell subset-specific differences in CD127 expression levels (as shown in Fig. 1) were preserved also in the presence of IL-7 (Fig. P9c, d, black crossed circles). To adjust for the partial decrease in CD127 signal in the presence of IL-7, we extended these data by calculating the fold difference in CD127 in PBS and Tofacitinib-treated samples (Fig. P9e, corresponding to Fig. 6i). Adjusting for differences in overall CD127 signal intensity showed that Tofacitinib was similarly potent in inhibiting IL-7-mediated CD127 reduction as it was in inhibiting IL-2- and IL-15-mediated CD127 reduction (Fig. P9e, corresponding to Fig. 6i).

12) It would be easier to interpret Fig 6A if the scales between the no activation and the CD3/CD28 stim were the same.

Response: We thank the reviewer for this comment. We have now adapted **Fig. 7a** accordingly in the revised manuscript.

13) It would be informative to indicate which groups significantly deviate from the healthy (dashed line) cohort in Fig S3C.

Response: We thank the reviewer for this comment. To address this point, we have now added an additional supplementary figure to our revised manuscript (**Fig. P10, corresponding to Supplementary Fig. 3f**). We have calculated and statistically tested the respective changes of mean receptor expression for each timepoint on CD4⁺ and CD8⁺ T cells following SARS-CoV-2 infection compared to the healthy control cohort. As expected, most significant and large changes compared to healthy steady state are observed during acute COVID-19 in individuals undergoing a severe disease course. Particularly in these patients, residual deviation from steady-state is still detectable at 6M-followup and in part at 12M-followup albeit with more attenuated log₂ fold changes.

14) Fig S9C may be more readily understood if rather than linking each stimulation condition at a given timepoint the authors display the results as a time course for each stimulation condition.

Response: We thank the reviewer for this comment and agree with the reviewer. Initially, we selected the representation of this figure to emphasize the different effects of cytokines at a given timepoint. To address the reviewer's point, we have now adapted this figure and added display items (**Fig. P12, corresponding to new Supplementary Fig. 13c–e**).

Summarily I like the manuscript and look forward to its eventual publication and hopefully the authors don't find these recommendations burdensome.

We greatly appreciate the reviewer's thoughtful and constructive feedback, which has contributed to strengthening the manuscript.

Reviewer 2:

In the manuscript, Ceglarek et al. performed a comprehensive analysis of gc cytokine receptor expression on each lymphocyte subset in healthy humans. Moreover, the authors show that CD25 expression levels are upregulated and remain high on T cells whereas CD127 levels downregulated and remain low for 6 months post SARS-CoV-2 infection and that these changes are associated with long-lasting alterations of IL-15 and to a lesser extent IL-7 levels in serum. Furthermore, they provide evidence showing that these homeostatic cytokines can directly upregulate and downregulate CD25 and CD127 expression on T cells, respectively, in vitro.

The manuscript is well written and can be an important addition to this research field. To consolidate their findings, I have the following comments and suggestions.

Response: We thank the reviewer for their generous and helpful comments. We have now addressed the reviewer's suggestions and comments in our revised manuscript as well as in our point-by-point responses to their specific points, as outlined below.

1) What are the functional consequences of prolonged CD25 upregulation and CD127 downregulation in human T lymphocytes? Can the authors add data showing the relationship between these changes and immunopathology in COVID-19 or other contexts?

Response: We thank the reviewer for raising this interesting question. Since these receptors govern homeostatic survival and activation of T cells, altered cytokine receptor levels are likely to result in functional differences in the T cell compartment, contributing to chronic inflammation and pathology. In line with this suggestion, we have now found that altered receptor expression correlates with prolonged T cell activation, both in SARS-CoV-2-specific and bystander T cell populations (**Fig. 4d, e**). To assess whether this dysregulated T cell phenotype correlates with chronic immunopathology, we compared receptor levels of individuals that clinically recovered from COVID-19 to those who reported long COVID (LC) symptoms for up to 12 months after COVID-19 (**Fig. P12a, b**). During acute COVID-19 and at 6-month (6M) follow-up there was a tendency for higher CD25 on total T cells in individuals that later developed LC persisting until 12M follow-up (**Fig. P12a**). This difference became more apparent at 12M follow-up where patients with persistent LC exhibited significantly higher CD25 levels on T cells. Furthermore, higher levels of CD25 were associated with more LC symptoms (**Fig. P12b**). Since we demonstrated that the magnitude of receptor alterations is dependent on initial disease severity (**Fig. 3c**), we compared CD25 levels on T cells of a subset of individuals who all experienced a mild disease course. Also in this cohort, patients with active LC at 12M follow-up showed elevated levels of CD25 in CD8⁺ T cells (**Fig. P12c**). To investigate whether this effect could be attributed to SARS-CoV-2-specific CD8⁺ T cells, we assessed CD25 expression in SARS-CoV-2-specific and influenza A virus (IAV)-specific CD8⁺ T cells in these individuals. CD25 levels were elevated on both SARS-CoV-2-specific and IAV-specific CD8⁺ T cells of LC individuals at 6-month follow-up. Furthermore, SARS-CoV-2-specific CD8⁺ T cells of LC patients with active symptoms showed higher CD25 levels compared to convalescent individuals at 12M follow-up. The fact that both SARS-CoV-2-specific and bystander CD8⁺ T cells showed differential CD25 expression at 6M follow-up, but only SARS-CoV-2-specific CD8⁺ T cells showed persistently elevated CD25 at 12M follow-up suggests a combined effect of both cytokine-mediated and potentially antigen-mediated signals. Elevation of CD25 in SARS-CoV-2-specific CD8⁺ T cells of LC individuals at 12M follow-up could indicate higher levels of persistent SARS-CoV-2 antigen, which has been reported by several studies (Stein, S. R. et al. *Nature* **612**, 2022; Peluso, M. J. et al. *Science Translational Medicine* **16**, 2024). Alternatively, the discrepancy of SARS-CoV-2- and IAV-

specific CD8⁺ T cells at 12M follow-up could be explained by the fact that SARS-CoV-2-specific CD8⁺ T cells were more recently primed by T cell receptor (TCR) signals, making them more prone for cytokine-mediated activation. In line with this suggestion, our *in vitro* stimulation assays (**Fig. 7**) showed a more vigorous cytokine response in cells pre-activated by TCR signals to those who received only cytokine. We have included and discussed these new data in our revised manuscript (**Supplementary Fig. 7** and lines 273-288).

2) Memory T cells are generally defined as those that can survive for long in the absence of cognate antigen recognition, and indeed in mice, LCMV Armstrong-specific memory CD4 and CD8 T lymphocytes can persist long after antigen deprivation. On the other hand, it has been reported that SARS-CoV-2 viral RNA can persist for 2 years in humans. Based on these observations, do the authors think that they are looking at "authentic" memory cells in 6/12-month follow-up samples of the present study? Or do dextramer(+) T cells on these time points represent persistently antigen-stimulated "effector" cells? This discussion should be included in the manuscript.

Response: We thank the reviewer for raising this interesting point, which we have now included in the discussion of our revised manuscript. Indeed, several studies reported that reservoirs of

SARS-CoV-2 RNA can persist in different human tissues (Stein, S. R. et al. *Nature* **612**, 2022; Peluso, M. J. et al. *Science Translational Medicine* **16**, 2024). However, the signatures of SARS-CoV-2–dextramer positive (CoV2–Dex⁺) cells in our data are consistent with memory cells rather than recently-stimulated effector cells. During acute COVID-19, CoV2–Dex⁺ CD8⁺ T cells expressed several effector markers, including HLA-DR and CXCR3, and showed high proliferation (Ki-67^{hi}). By contrast, expression of these markers was significantly decreased at 6M and 12M follow-up. Likewise, expression of transcription factors mediating cytotoxicity and effector responses, such as eomesodermin and T-bet, was most prominent during acute infection and decreased subsequently. In addition to extending the discussion, we have now also included these data in our revised manuscript (**Fig. P13, corresponding to Supplementary Fig. 5b**).

3) Fig 2D - G: Taking it into account that CoV2-Dex(+) T cells have the same or similar TCR specificity, what does the small "Ki67+ CD38+" subpopulation that accounts for CD25 upregulation on day 3 represent?

Response: We hypothesize that this subset represents a population of recent emigrants from secondary lymphoid organs (SLOs) still bearing the activation signature present in this anatomical location. In line with this suggestion, the overall frequency of CoV-2–Dex⁺ CD8⁺ T cells transiently decreases on day 3 followed by a strong numeric increase on subsequent sampling timepoints (**Fig. 2b**), indicating homing to SLOs and T cell activation, followed by recirculation. As CD25 upregulation following TCR stimulation seems to be very transient and short-lived in humans, high levels of CD25 on antigen-specific CD8⁺ T cells are most likely restricted to early T cell priming phases predominantly occurring in SLOs. Whether the fraction of CD25^{hi} cells emigrating from SLOs on day 3 represents cells with a certain TCR affinity is presently unclear and would require paired blood and SLO samples to conclusively address.

4) Fig 3F and G: The authors seem to consider CoV2-Dex(-) and CoV2-Dex(+) populations as bystander-activated and cognate antigen-stimulated cells, respectively (e.g., lines 212, 303, 370, etc.), although not explicitly stated. This is inaccurate or at least confusing because CoV2-Dex(-) T cells may contain cells that are specific for other SARS-CoV-2 antigens than those listed in Table S3. Interpretation on CoV2-Dex(-) T cells should be refined throughout the manuscript.

Response: We agree with the reviewer. Initially we reasoned that the frequency of SARS-CoV-2-reactive T cells within the CoV2–Dex⁻ CD8⁺ T cell compartment was unlikely to be sufficiently large to significantly skew the expression levels of IL-2R and IL-7R in the entire polyclonal T cell population. Particularly in the memory phase, the median frequencies of SARS-CoV-2-reactive CD8⁺ T cells were estimated to range between 0.2% and 0.5%, based on different *in vitro* peptide pool stimulation assays (Cohen et al. *Cell Rep. Med* **2** 2021; Dan et al. *Science* **371**, 2021). Nevertheless, we do agree with the reviewer that with the data

presented in Fig. 2F, G, we cannot formally exclude that receptor expression levels on CoV2–Dex⁻ cells are truly caused by TCR-independent cytokine signals and are not driven by SARS-CoV-2-specific T cells reactive to epitopes not covered by our CoV2–dextramer reagents. We have now addressed this, by performing another longitudinal profiling experiment of common gamma chain (γ c; also termed CD132) receptor subunits on a subset of our infection cohort (**Fig. P14; corresponding to new Fig. 4**). We included different dextramers carrying different influenza A virus (IAV)- and Epstein-Barr virus (EBV)-derived peptides to identify IAV- and EBV-specific bystander T cells and investigate whether signs of EBV reactivation can be detected in the post-acute infection stage (**Fig. P14a, b**). CoV-2-specific CD8⁺ T cells showed CD25 and CD122 upregulation and CD127 downregulation. We observed the same trends, but in attenuated fashion, on both IAV- and EBV-specific CD8⁺ T cells (**Fig. P14c, d**). Thus, both IAV- and EBV-specific CD8⁺ T cells showed signs of activation, marked by activation-associated molecules, such as CD38, HLA-DR, granzyme B (GzB) and Ki-67 (**Figs. P14e, and P3**). As expected, the most prominent shift toward highly activated clusters was observed in SARS-CoV-2-specific T cells during acute COVID-19 (**Figs. P14e and P15b**). Similarly, EBV- and IAV-specific CD8⁺ T cells adopted an activated phenotype, which was associated with high expression of CD122, a moderate increase in CD25 and downregulation of CD127 (**Fig. P14f**). Since EBV-specific cells could receive antigenic signals by latent viral antigens, we focused particularly on IAV-specific cells. On IAV-specific CD8⁺ T cells, upregulation of CD122 and downregulation of CD127 significantly correlated with the cells' activation status across all sampling timepoints, suggesting that aberrant expression of these homeostatic receptors correlated with prolonged bystander T cell activation (**Fig. P14g–i**). We have now included these data in the revised manuscript as Fig. 4 and adapted the corresponding results and discussion section accordingly (lines 242–266).

corresponding cells in healthy controls ($n = 13$). In **b–d**, HLA restriction prevented use of certain dextramers in all subjects. **e** UMAP projection of CoV2-, IAV- and EBV-specific CD8⁺ T cells in healthy individuals and COVID-19 patients, clustered based on activation markers *or* grouped by indicated virus specificities. **f** Heatmap depicting mean expression of indicated receptor subunits in different clusters of CoV2-, EBV- and IAV-specific CD8⁺ T cells. **g** Correlation between indicated receptor subunits and markers on IAV-specific CD8⁺ T cells. **h** Correlation of HLA-DR and CD122 and CD127 expression on IAV-specific CD8⁺ T cells, depicted as overall trend (black line), as well as data points during acute infection (magenta; $n = 17$), and at 6M (blue; $n = 17$) and 12M follow-up (green; $n = 17$). **i** HLA-DR, T-bet and Ki-67 expression on IAV-specific CD8⁺ T cells in COVID-19 patients ($n = 17$) at indicated timepoints.

Figure P15 (corresponding to Supplementary Fig. 6). Characterization of SARS-CoV-2-, influenza A virus (IAV)-, and Epstein-Barr virus (EBV)-specific CD8⁺ T cells during and following COVID-19. a Frequency of different memory T cell subsets among EBV- ($n = 10$) and IAV-specific ($n = 25$) CD8⁺ T cells at 12M follow-up. **b** Density UMAP of SARS-CoV-2-, IAV-, and EBV-specific CD8⁺ T cells across indicated timepoints. **c** UMAP projections of SARS-CoV-2-, IAV-, and EBV-specific CD8⁺ T cells from COVID-19 patients during acute COVID-19, and at 6M and 12M follow-up ($n = 26$) or healthy individuals ($n = 13$). Cells are colored by the scaled expression of indicated markers. **d** Heatmap depicting mean scaled marker expression of cells shown in (D) split by each identified cell cluster.

5) Fig 5D: The authors demonstrate that IL-7 strongly downregulates CD127 levels on T cells. Can they rule out the possibility that the binding of IL-7 to CD127 inhibited that of anti-CD127 mAb to CD127? Related to this point, Tofacitinib only minimally inhibited IL-7-induced CD127 downregulation whereas IL-2/15-triggered CD127 decrement was almost completely blocked by the same treatment (Fig 5H), suggesting that apparent CD127 downregulation in the presence of IL-7 might not be attributed to the effect of the same cytokine.

Response: We thank the reviewer for pointing this out. Previously we displayed linear geometric mean fluorescence intensities (gMFIs) of CD127 expression following treatment

with the Janus kinase (JAK) inhibitor Tofacitinib and cytokine stimulation. Following IL-7 stimulation, both PBS and JAK inhibitor-treated samples showed low CD127 levels. These data could be due to rapid internalization and downregulation upon IL-7 signaling. Or the data could be the result of a technical issue due to partial obstruction of the relevant epitope by recombinant IL-7. To address this latter point, we preincubated peripheral blood mononuclear cells (PBMCs) with IL-7 on ice for 20 min prior to staining with the surface staining mix, which also contained IL-7. Indeed, the presence of IL-7 decreased the signals of two different anti-CD127 antibody clones, namely hIL-7R-M21 and A019D5 (**Fig. P16a, b**). However, the residual CD127 staining signals were still significantly higher than the corresponding isotype controls. Furthermore, cell subset-specific differences in CD127 expression levels (as shown in **Fig. 1**) were preserved also in the presence of IL-7 (**Fig. P16c, d**, black crossed circles). To adjust for the partial decrease in CD127 signal in the presence of IL-7, we extended these data by calculating the fold difference in CD127 in PBS and Tofacitinib-treated samples (**Fig. P16e, corresponding to Fig. 6i**). Adjusting for differences in overall CD127 signal intensity showed that Tofacitinib was equally potent in inhibiting IL-7-mediated CD127 reduction as it was in inhibiting IL-2 and IL-15 mediated CD127 modulation.

6) Fig 5G: How did the authors analyze total (i.e., surface and intracellular) CD25 expression levels?

Response: We thank the reviewer for this comment and apologize for not having provided this information in the previous materials and methods. To determine total CD25 expression levels, we selected an anti-CD25 clone (M-A251) capable of binding to the fixed and native epitope of CD25. Following *in vitro* stimulation, we split each condition into two samples. To determine total CD25 levels, we fixed and permeabilized one sample using a commercially available fixation and permeabilization kit (eBioscience FoxP3/transcription factor staining buffer) and subsequently stained both surface and intracellular CD25. The other sample was stained for surface CD25 using the same fluorescently labeled mAb. To assure equal background autofluorescence, also samples stained for surface CD25 were subjected to the same fixation protocol after surface staining. We have now included a description of the staining protocol in the material and methods of our revised manuscript (lines 548-553)

7) The subheading "TCR signals without systemic inflammation" (line 124) is overstated. What they performed in this section is kinetic analysis of gc cytokine receptor expression in SARS-CoV-2 booster vaccination.

Response: In agreement with the reviewer we have now adapted the subheading in our revised manuscript, which now reads "SARS-CoV-2 booster vaccination induces transient IL-2R and IL-7R changes" (line 127).

8) Although the authors state "Following immunization CD25 induction precedes T cell activation markers and proliferation" (line 143), this description is incorrect because CD25 induction coincides with rather than precedes T cell proliferation on day 3 (Fig 2).

Response: We thank the reviewer for noticing this, as the subheading should have read: "Upon immunization CD25 precedes T cell activation markers and is associated with proliferation". We have now corrected this subheading in our revised manuscript (line 146).

9) What data do the authors refer to by "Paired analysis of CD25 levels in COVID-19 patients" (line 180)?

Response: The full cohort as depicted in Fig. 1A contains individuals infected with SARS-CoV-2 sampled during acute COVID-19 ($n = 64$), and at 6M ($n = 69$) and 12M follow-up ($n = 66$). Thus, certain individuals were lost to follow-up, or no blood sample was available for a given timepoint. Certain analyses, such as those depicted in Fig. 3e–g and in Supplementary Figs. 3 and 4, were conducted in unpaired fashion, comparing receptor expression or cell type frequencies on a per-group basis. Line 180 refers to the data depicted in Fig. 3c, which represent a fully paired dataset, only including patients of which data of all three timepoints were available. This allowed us to perform a paired comparison for each individual across time. Thanks to the reviewer's astute comment, we noticed that the sample size given in the legend of Fig. 3c mistakenly indicated the size of the full cohort, not of the paired subset, which we have corrected now in our revised manuscript.

10) The subheading "IL-7 and IL-15 mediate sustained changes of IL-2R and IL-7R in the absence of TCR activation" (line 231) is overstated because at least in this section, the authors just showed correlation but not causal association among IL-7/15 concentrations, CD25 levels, and T cell counts. Also, they cannot conclude "in the absence of TCR activation" in this section.

Response: We rephrased this subheading in our revised manuscript, which now reads: "Sustained changes in IL-2R and IL-7R abundance after acute SARS-CoV-2 infection".

11) The sample size (n) is not specified in the legend of Fig 6C.

Response: We thank the reviewer for picking up on this and have now included this information in our revised manuscript.

“Dysregulation of homeostatic cytokine receptors drives prolonged T cell activation following acute viral infection in humans” by Laura Ceglarek, Patrick Taeschler, Alp Inci, Yves Zurbuchen, Sarah Adamo, Carlo Cervia-Hasler, Miro E. Raeber and Onur Boyman

We thank the reviewers and editors for their positive assessment of our revised manuscript. Below we outline the minor changes we implemented according to the reviewers’ suggestions.

Reviewer 1

Summary:

The authors have convincingly provided data and rationale to address my previous comments, so I am happy to recommend the article for acceptance.

We thank the reviewer for their positive assessment of our revised manuscript.

Specific Points:

Minor note: I think that the data callout in line 254 is meant to be (Supplementary Fig. 6b) rather than (Supplementary Fig. 6e) though I suspect this can be fixed during the editorial stage prior to publication.

We thank the reviewer for noticing this typo and have corrected it in the manuscript.

Reviewer 2

Summary:

This reviewer thanks the authors for making substantial efforts on this revision. I have two minor comments as follows. Otherwise all my concerns have been comprehensively and satisfactorily addressed, and I recommend acceptance of this important paper for publication.

We thank the reviewer for their positive assessment of our revised manuscript.

Specific Points:

1) Related to my previous comment #5: While it is now clear that IL-7 can downregulate CD127 via JAK (Fig. 6i), it seems to be also true that IL-7-induced decrement of the CD127 MFI in Fig. 6d is at least in part due to obstruction of the relevant epitope by the same cytokine. I suggest to include a brief description of this issue in the main text.

Response: We have now included this point in the revised text (lines 359-361).

2) Related to my previous comment #9: In Fig. 3c, I suggest adding lines connecting data at three time points for each sample (as the authors did in Fig. S4e and S5) for readability of the sentence in line 186 (Paired analysis of CD25 levels...).

Response: We have adapted the figure accordingly, adding connecting lines for each paired sample (Fig. P1).